# MULTI-LEVEL REGRESSION FOR NONLINEAR CONTEXTUAL BANDITS AND RL: SECOND-ORDER AND HORIZON-FREE REGRET BOUNDS

## ABSTRACT

Recent works have established second-order regret bounds for nonlinear contextual bandits. However, these results exhibit a suboptimal dependence on the complexity of the function class. To close this gap, we propose a novel algorithm featuring a multi-level regression structure. This method partitions data by their uncertainty and variance, then performs separate regressions on each level, enabling adaptive, instance-dependent learning. Our method achieves a tight second-order regret bound of $\widetilde{O}\Big(\sqrt{d_{\mathcal{F}} \log N_{\mathcal{F}} \sum_{t \in [T]} \sigma_t^2} + R d_{\mathcal{F}} \log N_{\mathcal{F}}\Big)$, which matches the theoretical lower bound. Here, $d_{\mathcal{F}}$ and $\log N_{\mathcal{F}}$ represent the Eluder dimension and log-covering number of the reward function class $\mathcal{F}$, $\sigma_t^2$ is the unknown variance of the reward at round $t$, and $R$ is the range of rewards. The proposed algorithm is computationally efficient assuming access to a regression oracle. We further extend our framework to model-based reinforcement learning, achieving a regret bound that is both second-order and horizon-free. The underlying multi-level regression technique is of independent interest and applicable to a broad range of online decision-making problems.

## 1 INTRODUCTION

In the realm of online decision-making problems, contextual bandits serve as a foundational model, where an agent interacts with the environment to learn and act optimally in the face of uncertainty. This paradigm is central to numerous real-world applications, including personalized recommendation systems (Li et al., 2010; Covington et al., 2016), dynamic pricing (Kleinberg & Leighton, 2003; Ferreira et al., 2018), and online advertising (Agarwal et al., 2014; Chapelle et al., 2014). A central goal in this field is to design algorithms with strong performance guarantees measured by regret—the difference in rewards between the algorithm's choices and those of an optimal policy. Although worst-case regret bounds have been well studied (Auer et al., 2002; Abbasi-Yadkori et al., 2011), the field has increasingly focused on developing more nuanced, instance-dependent guarantees (Zhou & Gu, 2022; Li & Sun, 2024; Huang et al., 2023). Second-order regret bounds, which incorporate the *unknown* variance of the rewards, are particularly valuable as they adapt to the problem's intrinsic statistical difficulty rather than relying on pessimistic worst-case guarantees.

Despite significant progress in linear contextual bandits (Zhao et al., 2023), a fundamental challenge has persisted in the setting of general function approximation, which is critical for capturing the complex relationships present in real-world scenarios. Current algorithms often suffer from a suboptimal dependence on the complexity of the reward function class, such as the Eluder dimension $d_{\mathcal{F}}$. For instance, the best-known algorithms from Pacchiano (2025); Jia et al. (2024) achieve regret bounds of the form $\widetilde{O}\Big(d_{\mathcal{F}} \sqrt{\log N_{\mathcal{F}} \sum_{t \in [T]} \sigma_t^2} + R d_{\mathcal{F}} \log N_{\mathcal{F}}\Big)$, which falls short of the theoretical lower bound by Jia et al. (2024) that suggests a $\sqrt{d_{\mathcal{F}}}$ dependency. While Wang et al. (2024b) achieve a $O(\sqrt{d_{\mathcal{P}}})$ regret bound, their algorithm requires a stronger realizability assumption: access to the full reward distribution. This discrepancy raises a crucial open question:

*Can we design an algorithm for nonlinear contextual bandits that achieves a minimax-optimal, second-order regret with the standard realizability assumption?*

Table 1: Regret bounds of algorithms for contextual bandits with *unknown* reward variances. Here, $d$ denotes the dimension for linear function approximation, $\mathcal{P}$ represents the reward distribution class, $\mathcal{F}$ is the the reward function class, $d_{\mathcal{F}}, d_{\mathcal{P}}$ are the Eluder dimension, $N_{\mathcal{F}}, N_{\mathcal{P}}$ are the covering number, $T$ is the number of rounds, $\sigma_t$ is the variance of the reward at round $t$, and $R$ is the range of rewards. $\widetilde{O}$ omits logarithmic terms.

| Algorithm | Function Type | Regret Bound | Computational Efficiency |
|---|---|---|---|
| SAVE (Zhao et al., 2023) | Linear | $\widetilde{O}\left(d\sqrt{\sum_{t\in[T]}\sigma_t^2} + Rd\right)$ | Yes |
| DistUCB (Wang et al., 2024b) | Nonlinear | $\widetilde{O}\left(\sqrt{d_{\mathcal{P}}\log N_{\mathcal{F}}\sum_{t\in[T]}\sigma_t^2} + Rd_{\mathcal{P}}\log N_{\mathcal{P}}\right)$ | Yes |
| Unknown-Variance SOOLS (Pacchiano, 2025) VarUCB (Jia et al., 2024) | Nonlinear | $\widetilde{O}\left(d_{\mathcal{F}}\sqrt{\log N_{\mathcal{F}}\sum_{t\in[T]}\sigma_t^2} + Rd_{\mathcal{F}}\log N_{\mathcal{F}}\right)$ | Yes |
| VACB (Ye et al., 2025) | Nonlinear | $\widetilde{O}\left(d_{\mathcal{F}}\sqrt{\log N_{\mathcal{F}}\sum_{t\in[T]}\sigma_t^2} + d_{\mathcal{F}}(\log N_{\mathcal{F}})^{3/4}\right)$ | No |
| UCB-MLR (Theorem 4.2) | Nonlinear | $\widetilde{O}\left(\sqrt{d_{\mathcal{F}}\log N_{\mathcal{F}}\sum_{t\in[T]}\sigma_t^2} + Rd_{\mathcal{F}}\log N_{\mathcal{F}}\right)$ | Yes |

We give an affirmative answer to this question by delving into the problem of nonlinear contextual bandits. Specifically, we consider the setting with *heteroscedastic* noise—where the variance of rewards changes over time—and, critically, we assume this variance is *unknown* to the agent, a common scenario in real-world applications. Our contributions are summarized as follows:

- We propose a novel Multi-Level Regression (MLR) structure, which significantly advances prior multi-layer algorithms inspired from Zhao et al. (2023). A key innovation lies in our data partitioning method, ADALEVEL, which leverages both uncertainty and variance rather than just uncertainty. By running separate regressions on each level, our algorithm learns in an adaptive and instance-dependent way, leading to a more accurate function estimate. The principles of this multi-level regression technique are broadly applicable and may be of independent interest for other online decision-making problems.

- Leveraging our new technique, we propose UCB-MLR, a novel algorithm for nonlinear contextual bandits. Through the use of a tighter Bernstein-style bound for nonlinear regression and a detailed analysis of estimation error at different levels, we theoretically establish a regret bound of $\widetilde{O}\left(\sqrt{d_{\mathcal{F}}\log N_{\mathcal{F}}\sum_{t\in[T]}\sigma_t^2} + Rd_{\mathcal{F}}\log N_{\mathcal{F}}\right)$. This result is significant because it is the first to match the second-order lower bound from Jia et al. (2024), effectively resolving a suboptimal dependency on $d_{\mathcal{F}}$. Our algorithm also achieves computational efficiency with access to a regression oracle.

- We further demonstrate the effectiveness and generality of our algorithmic framework by applying it to model-based Reinforcement Learning (RL), where an agent learns to act optimally by building a model of the environment. Our proposed algorithm, ML-VTR, is the first to achieve a regret bound of $\widetilde{O}\left(\sqrt{d_{\mathcal{F}}\log N_{\mathcal{F}}\,\mathrm{Var}_K^*} + d_{\mathcal{F}}\log N_{\mathcal{F}}\right)$ for Markov Decision Processes (MDPs) with general function approximation. This result is notable because it is simultaneously second-order, horizon-free, and computationally efficient. As a special case, it reduces to $\widetilde{O}\left(d\sqrt{\mathrm{Var}_K^*} + d^2\right)$ for linear mixture MDPs. This bound matches the state-of-the-art from Zhao et al. (2023), suggesting that our novel algorithm and fine-grained analysis are effective for a wide range of general RL problems.

For a comprehensive comparison with state-of-the-art results, we summarize the regrets in Table 1 for contextual bandits and Table 2 for RL.

**Notations** Let $[n] := \{1, 2, \ldots, n\}$, $\overline{[n]} := \{0, 1, \ldots, n\}$, and $X_{\mathcal{I}} := \{X_i\}_{i\in\mathcal{I}}$. Denote $\min_{x\in\mathcal{X}}\{c, f(x)\} := \min\{c, \min_{x\in\mathcal{X}} f(x)\}$ for short. Denote the $\epsilon$-covering number of $\mathcal{F}$ w.r.t. $\ell_\infty$-norm as $N_{\mathcal{F}}(\epsilon)$. $\widetilde{O}(\cdot)$ omits logarithmic terms in $O(\cdot)$.

Table 2: Regret bounds of algorithms for model-based RL that achieve *instance-dependent* and *horizon-free*. Here, $d$ denotes the dimension for linear function approximation, $\mathcal{P}$ represents the transition model class, $\mathcal{F}$ is the the function class induced by $\mathcal{P}$, $d_\mathcal{F}, d_\mathcal{P}$ are the Eluder dimension, $N_\mathcal{F}, N_\mathcal{P}$ are the covering number, quantity $\mathrm{Var}_K^*$ defined in (5.3) is the total variance of the optimal value functions, and $\mathcal{Q}^*$ is a higher-order moments quantity defined in Huang et al. (2024). $\widetilde{O}$ omits logarithmic terms.

| Algorithm | Function Type | Regret Bound | Computational Efficiency |
|---|---|---|---|
| UCRL-AVE (Zhao et al., 2023) | Linear | $\widetilde{O}\big(d\sqrt{\mathrm{Var}_K^*} + d^2\big)$ | Yes |
| UCRL-WVTR (Huang et al., 2024) | Nonlinear | $\widetilde{O}\big(\sqrt{d_\mathcal{F} \log N_\mathcal{F} \mathcal{Q}^*} + d_\mathcal{F} \log N_\mathcal{F}\big)$ | Yes |
| O-MBRL (Wang et al., 2025) | Nonlinear | $\widetilde{O}\big(\sqrt{d_\mathcal{P} \log N_\mathcal{P} \mathrm{Var}_K^*} + d_\mathcal{P} \log N_\mathcal{P}\big)$ | No |
| ML-VTR (Theorem 5.1) | Nonlinear | $\widetilde{O}\big(\sqrt{d_\mathcal{F} \log N_\mathcal{F} \mathrm{Var}_K^*} + d_\mathcal{F} \log N_\mathcal{F}\big)$ | Yes |

## 2 RELATED WORK

**Second-Order Regret in Nonlinear Contextual Bandits**  Designing algorithms with second-order regret has become a central theme in contextual bandits literature. While the linear setting is well-understood (Zhao et al., 2023), the nonlinear setting with *unknown* variances presents substantially greater challenges, revealing a distinct gap to statistical optimality.

Several attempts, such as Unknown-Variance SOOLS (Pacchiano, 2025) and VarUCB (Jia et al., 2024), have been made to generalize the multi-layer technique developed in (Zhao et al., 2023) to nonlinear settings. Furthermore, VACB (Ye et al., 2025) utilizes Catoni estimator to handle the heavy-tailedness of noise, removing the $R$ dependence on the lower order. However, due to the intrinsic difficulty caused by nonlinear structure, they only obtain a regret that is suboptimal on the function complexity, thereby leaving a gap to optimality. A different line of work (Foster et al., 2018; Wang et al., 2024b;a), exemplified by DistUCB (Wang et al., 2024b), pursue variance-adaptivity using MLE for the full reward distribution. However, this distributional approach requires the stronger and often impractical modeling assumption that the entire reward distribution—not just the expected reward—is realizable by the model class. Our multi-level regression framework, by contrast, operates under the standard, less restrictive realizability assumption.

**Instance-dependent and Horizon-free Regret in Model-based RL**  The principles of instance-dependent learning are also paramount in the more complex domain of RL, where the additional challenges of long-planning horizons must be addressed. A key goal in modern RL theory is to develop algorithms that are not only second-order but also horizon-free, meaning their regret bounds scale at most polylogarithmicly with the planning horizon $H$ (Jiang & Agarwal, 2018). For MDPs with linear function approximation, also known as linear mixture MDPs, Zhao et al. (2023) provide an efficient, second-order and horizon-free algorithm. However, extending these successes to general function approximation presents significant challenges.

To name a few, Huang et al. (2024) made the first attempt to propose an algorithm, UCRL-WVTR, using weighted value-targeted regression for estimating the model and achieves an instance-dependent and horizon-free regret. Despite worst-case optimal when specialized to linear mixture MDPs, their regret bound has a suboptimal dependence on the higher-order moments of the optimal value functions. Conversely, O-MBRL (Wang et al., 2025) extends DistUCB to RL and achieves a tight, second-order and horizon-free statistical guarantee. However, it is generally computationally intractable and requires the stronger assumption of access to the full distribution.

## 3 PRELIMINARIES

**Nonlinear Contextual Bandits**  We consider a $T$-round contextual bandit problem. At each round $t \in [T]$, the environment provides a candidate decision set $\mathcal{X}_t \subseteq \mathcal{X}$. This framework includes the classic contextual bandit setting given context $z_t$ and action set $\mathcal{A}$, by setting $\mathcal{X}_t = \{z_t\} \times \mathcal{A}$. The

agent selects an action $x_t \in \mathcal{X}_t$ and receives a reward $y_t = f_*(x_t) + \varepsilon_t$. We assume $y_t \in [0, R]$,

$$\mathbb{E}[\varepsilon_t | x_t] = 0, \quad \mathrm{Var}[\varepsilon_t | x_t] = \mathrm{Var}[y_t | x_t] = \sigma_t^2 \le \sigma^2.$$

To enable the utilization of a priori unknown variance information, we make Assumption 3.2, which is also adopted by Ye et al. (2025).

**Assumption 3.1** (Realizability). We are given access to a function class $\mathcal{F}$ such that $f_* \in \mathcal{F}$.

**Assumption 3.2.** We are given access to a function class $\mathcal{G}$ and constant $c_v > 0$ such that $g_* \in \mathcal{G}$, and for all rounds $t \in [T]$,

$$\mathbb{E}[y_t^2 | x_t] = g_*(x_t), \quad \mathrm{Var}[y_t^2 | x_t] \le c_v^2 R^2 \cdot \mathrm{Var}[y_t | x_t] = c_v^2 R^2 \sigma_t^2.$$

We use the standard Eluder dimension and covering number to measure the complexity of $\mathcal{F}$. Recall the definition of Eluder dimension (Russo & Van Roy, 2013):

**Definition 3.3** (Eluder Dimension). Let $\mathcal{F}$ be a function class defined on $\mathcal{X}$ and $\epsilon > 0$. The Eluder dimension $\dim_{\mathcal{F}}(\epsilon)$ of $\mathcal{F}$ is the length of the longest sequence $x_{[n]} \subseteq \mathcal{X}$ such that for some $\epsilon' \ge \epsilon$, for all $t \le n$, $x_t$ is $\epsilon'$-independent of $x_{[t-1]}$ given $\mathcal{F}$. That is, there exists $f, f' \in \mathcal{F}$ such that

$$\sum_{s \in [t-1]} [f(x_s) - f'(x_s)]^2 \le \epsilon'^2 \text{ while } |f(x_t) - f'(x_t)| > \epsilon'.$$

We also use the notation $d_{\mathcal{F}} := \dim_{\mathcal{F}}(\epsilon)$ and $N_{\mathcal{F}} := N_{\mathcal{F}}(\epsilon)$ for short when $\epsilon$ is clear from the context. Let $\lambda > 0$. We quantify uncertainty of $x$ given dataset $x_{[t-1]}$ and wights $w_{[t-1]}$ w.r.t. $\mathcal{F}$ as:

$$D_{\mathcal{F}}(x; x_{[t-1]}, w_{[t-1]}) := \sup_{f_1, f_2 \in \mathcal{F}} \frac{(f_1(x) - f_2(x))^2}{\sum_{s \in [t-1]} w_s^2 (f_1(x_s) - f_2(x_s))^2 + \lambda}. \tag{3.1}$$

**MDPs with General Function Approximation**  We consider episodic MDPs defined by a tuple $(\mathcal{S}, \mathcal{A}, H, \mathbb{P}, \{r_h\}_{h \in [H]})$. Here, $\mathcal{S}$ and $\mathcal{A}$ are the state space and action spaces, $H$ is the planning horizon, $\mathbb{P} : \mathcal{S} \times \mathcal{A} \to \Delta(\mathcal{S})$ is the transition dynamics, $r_h : \mathcal{S} \times \mathcal{A} \to \mathbb{R}$ is the $h$-th step reward function known to the agents[1]. We assume a bounded reward setting where $\sum_{h=1}^H r_h(s_h, a_h) \le 1$ for any trajectory. We use a deterministic policy throughout this paper, which is a collection of $H$ mappings from the state space to the action space, denoted as $\pi = \{\pi_h : \mathcal{S} \to \mathcal{A}\}_{h \in [H]}$. For any state-action pair $(s, a) \in \mathcal{S} \times \mathcal{A}$, the action value function $Q_h^\pi(s, a)$ and the (state) value function $V_h^\pi(s)$ are defined as:

$$Q_h^\pi(s, a) := \mathbb{E}\Big[ \sum_{h'=h}^H r(s_{h'}, a_{h'}) \Big| s_h = s, a_h = a \Big], \quad V_h^\pi(s) := Q_h^\pi(s, \pi_h(s)),$$

where the expectation is taken w.r.t. the transition kernel $\mathbb{P}$ and the agent's policy $\pi$. We denote the optimal value functions as $V_h^*(s) := \sup_\pi V_h^\pi(s)$ and $Q_h^*(s, a) := \sup_\pi Q_h^\pi(s, a)$. For simplicity, we introduce the following shorthands. Let $\mathcal{V}$ be the set of all value functions $V : \mathcal{S} \to [0, 1]$. For any $V \in \mathcal{V}$, we denote the conditional expectation and variance of $V$ as

$$[\mathbb{P}V](s, a) := \mathbb{E}_{s' \sim \mathbb{P}(\cdot | s, a)}[V(s')], \quad [\mathbb{V}V](s, a) := [\mathbb{P}V^2](s, a) - [\mathbb{P}V]^2(s, a)$$

Our objective is to design efficient algorithms that minimize the $K$-episode regret, defined as

$$\mathrm{Regret}(K) := \sum_{k=1}^K \big( V_1^*(s_1^k) - V_1^{\pi^k}(s_1^k) \big).$$

To solve problems of large state spaces, we consider MDPs with general function approximation. We adopt the following assumptions to accurately estimate the variance of value functions, which is reasonable since a small variance of a next-state value function often indicates more deterministic transitions, thus suggesting a small variance for the squared next-state value function.

**Assumption 3.4** (Realizability). Let $\mathcal{P}$ be a general function class consisting of transition kernels that map state-action pairs to measures over $\mathcal{S}$. We assume the MDP's transition model $\mathbb{P} \in \mathcal{P}$.

---

[1]We consider deterministic rewards since our result can be easily generalized to the unknown-reward cases.

---

**Algorithm 1** UCB-MLR

---

**Require:** $\alpha, \widetilde{\alpha}, \gamma, \widetilde{\gamma}, L = \lceil \log_2 \frac{R}{\alpha} \rceil, \widetilde{L} = \lceil \log_2 \frac{R^2}{\widetilde{\alpha}} \rceil, \{\beta_{t,l}\}_{t \geq 1, l \in [L]}, \{\widetilde{\beta}_{t,\ell}\}_{t \geq 1, \ell \in [\widetilde{L}]}$

1: $\Psi_{1,l} \leftarrow \varnothing$ for $l \in \overline{[L]}$, $\widetilde{\Psi}_{1,\ell} \leftarrow \varnothing$ for $\ell \in \overline{[\widetilde{L}]}$
2: $\widehat{f}_{1,l} \leftarrow 0$ for $l \in [L]$, $\widehat{g}_{1,\ell} \leftarrow 0$ for $\ell \in [\widetilde{L}]$
3: **for** $t = 1, \dots, T$ **do**
4:     Observe $\mathcal{X}_t$
5:     Choose $x_t \leftarrow \operatorname{argmax}_{x \in \mathcal{X}_t} \min_{l \in [L]} \left( \widehat{f}_{t,l}(x) + \min\{R, \beta_{t,l} D_{t,l}(x)\} \right)$, receive $y_t$
6:     Update $\bar{\sigma}_t$ according to (4.3).
7:     Set $l_t, w_t \leftarrow \text{ADALEVEL}\left( \{D_{t,l}(x_t)\}_{l \in [L]}, \bar{\sigma}_t, \alpha, \gamma \right)$
8:     Set $\ell_t, \widetilde{w}_t \leftarrow \text{ADALEVEL}\left( \{\widetilde{D}_{t,\ell}(x_t)\}_{\ell \in [\widetilde{L}]}, c_v \bar{\sigma}_t, \widetilde{\alpha}, \widetilde{\gamma} \right)$
9:     Update $\Psi_{t+1,l_t} \leftarrow \Psi_{t,l_t} \cup \{t\}$, $\Psi_{t+1,l} \leftarrow \Psi_{t,l}$ for $l \in \overline{[L]}, l \neq l_t$
10:    Update $\widetilde{\Psi}_{t+1,\ell_t} \leftarrow \widetilde{\Psi}_{t,\ell_t} \cup \{t\}$, $\widetilde{\Psi}_{t+1,\ell} \leftarrow \widetilde{\Psi}_{t,\ell}$ for $\ell \in \overline{[\widetilde{L}]}, \ell \neq \ell_t$
11:    Update $\widehat{f}_{t+1,l}$ for $l \in [L]$, $\widehat{g}_{t+1,\ell}$ for $\ell \in [\widetilde{L}]$ according to (4.1), (4.2)
12: **end for**

---

**Algorithm 2** ADALEVEL

---

**Require:** $\{D_{t,l}\}_{l \in [L]}, \bar{\sigma}_t, \alpha, \gamma$
**Ensure:** Level $l_t$, weight $w_t$
1: Set $l_t \leftarrow \max\{l \in [L] : \gamma D_{t,l} > 2^l \alpha\}$
2: **if** $l_t = -\infty$ **then**
3:     Update $l_t \leftarrow 0$
4: **else**
5:     **if** $\bar{\sigma}_t \leq 2^{l_t} \alpha$ **then**
6:         Set $w_t \leftarrow \frac{2^{l_t} \alpha}{\gamma D_{t,l_t}}$
7:     **else**
8:         Update $l_t \leftarrow \min\{l \in [L], l > l_t : \bar{\sigma}_t \leq 2^l \alpha\}$
9:         Set $w_t \leftarrow 1$
10:    **end if**
11: **end if**

---

**Assumption 3.5.** There exists a constant $c_v > 0$ such that for all steps $(k, h) \in [K] \times [H]$ and all $V_{h+1} \in \mathcal{V}$, the following holds:

$$[\mathbb{V} V_{h+1}^2](s_h^k, a_h^k) \leq c_v^2 [\mathbb{V} V_{h+1}](s_h^k, a_h^k).$$

We use the covering number and Eluder dimension to measure the complexity of the function class $\mathcal{F}$, which is induced from the model class $\mathcal{P}$. $\mathcal{F}$ is generally smaller than $\mathcal{P}$, since we only require the expectation instead of the distribution information.

$$\mathcal{F} := \{f : \mathcal{S} \times \mathcal{A} \times \mathcal{V} \to \mathbb{R} \mid \exists \mathbb{P} \in \mathcal{P}, f(s_h^k, a_h^k, V_{h+1}) = [\mathbb{P} V_{h+1}](s_h^k, a_h^k)\},$$

# 4 MULTI-LEVEL REGRESSION FOR CONTEXTUAL BANDITS

In this section, we propose a new algorithm for nonlinear contextual bandits, UCB-MLR, which is formally presented in Algorithm 1. We introduce the notation $D_{t,l}(x) := D_{\mathcal{F}}(x; x_{\Psi_{t,l}}, w_{\Psi_{t,l}})$ and $\widetilde{D}_{t,\ell}(x) := D_{\mathcal{G}}(x; x_{\widetilde{\Psi}_{t,\ell}}, w_{\widetilde{\Psi}_{t,\ell}})$ for conciseness. We first outline the high-level idea, then analyze the computational complexity and regret bound.

## 4.1 ALGORITHM DESCRIPTION

UCB-MLR improves upon the multi-layer structure proposed by Zhao et al. (2023). Their approach partitions data into $L + 1$ layers based on uncertainty, performs regressions within each layer $l \in [L]$, and combine $L$ results to form a more accurate estimate of the reward function. In contrast, our

leveling algorithm, ADALEVEL, partitions date using both uncertainty and variance. We highlight the primary enhancements of UCB-MLR in as follows:

**Adaptive Leveling**   In Line 7 of Algorithm 1, ADALEVEL adaptively chooses the level $l_t$ for each data point $x_t$ at round $t$, as detailed in Algorithm 2. This selection, based on its uncertainty within each level $\{D_{t,l}(x_t)\}_{l \in [L]}$ and the estimated variance $\bar{\sigma}_t^2$, leverages the concentration inequality in Lemma 4.3 to reduce the estimation error of reward function $f_*$.

We use $\Psi_{t+1,l}$ to denote the index set of all date partitioned into level $l \in \overline{[L]}$ up to time $t$. The detailed properties of ADALEVEL are listed in Property 1. In general, for all $t \in [T]$ such that $t \in \Psi_{T+1,l}$ with $l \in [L]$, we set weight

$$w_t = \min\left\{1, \frac{2^l \alpha}{\gamma D_{t,l}(x_t)}\right\}.$$

This is done to avoid a sharp change in uncertainty between adjacent levels. Consequently, we have:

$$w_t D_{t,l} \le 2^l \alpha / \gamma, \quad w_t \bar{\sigma}_t \le 2^l \alpha,$$

where $\alpha$ and $\gamma$ are prespecified parameters. This ensures that the data at level $l$ have roughly the same uncertainty and variance, both on the order of $2^l \alpha$. We use ADALEVEL similarly to construct $\{\widetilde{\Psi}_{T,\ell}\}_{\ell \in \overline{[\widetilde{L}]}}$ for estimating the squared-reward function $g_*$.

**Multi-Level Regression and Upper Confidence Bound (UCB)**   At round $t$, after updating $\{\Psi_{t+1,l}\}_{l \in \overline{[L]}}$ and $\{\Psi_{t+1,\ell}\}_{\ell \in \overline{[\widetilde{L}]}}$, we utilize weighted least squares regression to estimate $f_*$ for level $l \in [L]$ and $g_*$ for level $\ell \in [\widetilde{L}]$:

$$\widehat{f}_{t+1,l} = \operatorname*{argmin}_{f \in \mathcal{F}} \sum_{s \in \Psi_{t+1,l}} w_s^2 (f(x_s) - y_s)^2, \tag{4.1}$$

$$\widehat{g}_{t+1,\ell} = \operatorname*{argmin}_{g \in \mathcal{F}} \sum_{s \in \widetilde{\Psi}_{t+1,\ell}} \widetilde{w}_s (g(x_s) - y_s^2)^2. \tag{4.2}$$

As shown in Line 5, for any $x \in \mathcal{X}$, we can construct $L$ high-probability UCBs for $f_*(x)$ and take their minimum to choose the action optimistically:

$$f_*(x) \le \min_{l \in [L]} \left(\widehat{f}_{t+1,l}(x) + \min\{R, \beta_{t+1,l} D_{t+1,l}(x)\}\right),$$

Similarly, we can set $\bar{\sigma}_t^2$ as the upper bound of $\sigma_t^2$:

$$\bar{\sigma}_t^2 = \min_{l \in [L], \ell \in [\widetilde{L}]} \left\{\sigma^2, \widehat{g}_{t,\ell}(x_t) - \widehat{f}_{t,l}^2(x_t) + R\min\{R, 2\beta_{t,l} D_{t,l}(x_t)\} + \min\{R^2, \widetilde{\beta}_{t,\ell} \widetilde{D}_{t,\ell}(x_t)\}\right\}. \tag{4.3}$$

According to Lemma 4.3, $\beta_{t,l} = \widetilde{O}(2^l \alpha \sqrt{\log N_{\mathcal{F}}})$ and $\widetilde{\beta}_{t,\ell} = \widetilde{O}(2^\ell \widetilde{\alpha} \sqrt{\log \mathcal{N}_{\mathcal{G}}})$.

**Computational Complexity**   We analyze the computational complexity of UCB-MLR, relying on a regression oracle defined in Assumption 4.1 for solving the weighted nonlinear least squares regression. By adopting the techniques from Li et al. (2023); Huang et al. (2024), we can leverage this oracle to compute the uncertainty $\mathcal{D}_{\mathcal{F}}$ defined in (3.1) through a binary search procedure, requiring only $\widetilde{O}(1)$ calls to the oracle.

**Assumption 4.1** (Regression Oracle). We assume access to a weighted least squares *regression oracle*, which takes a function class $\mathcal{F}$ and $t$ weighted examples $\{(X_s, w_s, Y_s)\}_{s \in [t]} \subseteq \mathcal{X} \times \mathbb{R}^+ \times \mathbb{R}$ as input. It then outputs the solution to the weighted least squares problem, $\widehat{f}$, within $\mathcal{R}$ time, where

$$\widehat{f} = \operatorname*{argmin}_{f \in \mathcal{F}} \sum_{s=1}^{t} w_s (f(X_s) - Y_s)^2.$$

We now consider the computation cost for a single round $t$. First, computing the UCB of $f_*(x_t)$ in Line 5 requires $\widetilde{O}(L\mathcal{R})$ time, as it involves calculating $\mathcal{D}_{t,l}$ for $l \in [L]$. To select the best action over the set $\mathcal{X}_t$, the algorithm must compute the UCB for at most $|\mathcal{X}|$ actions. Next, the estimated variance $\bar{\sigma}_t^2$ in (4.3) can be computed within $\widetilde{O}((L + \widetilde{L})\mathcal{R})$ time. And ADALEVEL takes $\widetilde{O}(L + \widetilde{L})$ time. Finally, it takes $(L + \widetilde{L})\mathcal{R}$ time to calculate the regression estimates $\widehat{f}_{t+1,l}$ in (4.1) for $l \in [L]$ and $\widehat{g}(t + 1, \ell)$ in (4.2). Therefore, the total computational cost of UCB-MLR is $\widetilde{O}(T|\mathcal{X}|\mathcal{R})$.

## 4.2 REGRET BOUND

**Theorem 4.2.** For contextual bandit with general function approximation as defined in Section 3, if the parameters in Algorithm 1 are set according to Section B, then with probability at least $1 - (L + \widetilde{L})\delta$, UCB-MLR achieves

$$\text{Regret}(T) = \widetilde{O}\Big(\sqrt{d_{\mathcal{F}} \log N_{\mathcal{F}} \sum\nolimits_{t \in [T]} \sigma_t^2} + \max\{1, C\} R d_{\mathcal{F}} \log N_{\mathcal{F}}\Big),$$

where $C = \max\{1, c_v\}\sqrt{\frac{d_{\mathcal{G}} \log N_{\mathcal{G}}}{d_{\mathcal{F}} \log N_{\mathcal{F}}}}$.

*Proof.* The proof uses a tighter Bernstein-style bound for the estimated function and a detailed analysis of the summation of bonuses within each level. See Section 4.3 for a proof sketch and Section B for a detailed proof. □

Our result matches the second-order lower bound established by Jia et al. (2024), therefore successfully eliminating the gap related to $d_{\mathcal{F}}$. We leave the removal of $C$ in the lower order term as an open problem for future work.

As a special case, for a $d$-dimensional linear contextual bandit, where $d_{\mathcal{F}}, \log N_{\mathcal{F}} = O(d)$ (Jia et al., 2024), our algorithm achieves a regret of $\widetilde{O}\Big(d\sqrt{\sum_{t \in [T]} \sigma_t^2} + Rd^2\Big)$. This matches the state-of-the-art result of Zhao et al. (2023) for the main term.

## 4.3 PROOF SKETCH

**Concentration of the Estimated Function**  Our primary effort is to establish a tight UCB for the true reward function $f_*$. This relies on the concentration inequality presented in Lemma 4.3.

**Lemma 4.3.** Let $\{X_t\}_{t \geq 1} \subseteq \mathcal{X}$ and $\{Y_t\}_{t \geq 1} \subseteq [0, R]$ be sequences of random elements, and let $\{w_t\}_{t \geq 1}$ be a sequence of weights. Let $f_* \in \mathcal{F}$ with function class $\mathcal{F} : \mathcal{X} \to [0, R]$. Suppose for all $s \in [t]$, $\mathbb{E}[Y_s | X_s] = f_*(X_t)$, $|w_s| \leq W$, and $w_s^2 \text{Var}[Y_s | X_s] \leq \sigma^2$. Let the estimated function be

$$\widehat{f}_{t+1} = \operatorname*{argmin}_{f \in \mathcal{F}} \sum_{s=1}^{t} w_s^2 (f(X_s) - Y_s)^2. \tag{4.4}$$

Then for any $\delta, \epsilon > 0$, with probability at least $1 - \delta$, we have for all $t \geq 1$,

$$\sum_{s=1}^{t} w_s^2 (\widehat{f}_{t+1}(X_s) - f_*(X_s))^2 \leq \beta_{t+1}^2 \text{ with}$$

$$\beta_{t+1} = 3\sqrt{\iota_t}\sigma + 2\iota_t R \min\Big\{1, \max_{s \in [t]} w_s^2 D_{\mathcal{F}}(X_s; X_{[s-1]}, w_{[s-1]})\Big\} + \sqrt{\lambda} + \sqrt{6W^2 R t \epsilon},$$

where $\iota_t = 16 \log \frac{2N_{\mathcal{F}}(\epsilon)t^2(\log(\sigma^2 W^2 R^2 t) + 2)(\log(W^2 R^2) + 2)}{\delta} = \widetilde{O}(\log N_{\mathcal{F}})$.

*Proof.* See Section A for a detailed proof. □

**Remark 4.4.** Lemma 4.3 improves upon the Bernstein-style bound for nonlinear regression from Huang et al. (2024) by tightening the term concerning uncertainty. Here, we denote $D_t := \max_{s \in [t]} D_{\mathcal{F}}(X_s; X_{[s-1]}, w_{[s-1]})$ for short. This implies the confidence radius $\beta_{t+1} = \widetilde{O}(\sigma\sqrt{\log N_{\mathcal{F}}} + RD_t \log N_{\mathcal{F}})$. Compared to the bound $\widetilde{O}\big(D_t\sqrt{\sum_{s \in [t]} \sigma_s^2 \log N_{\mathcal{F}}} + RD_t \log N_{\mathcal{F}}\big)$ used in previous multi-layer algorithms (Pacchiano, 2025; Jia et al., 2024), our result improves the first term by a factor of $\sqrt{d_{\mathcal{F}}}$ when the reward variances are roughly equal, since $D_t$ is of order $\sqrt{d_{\mathcal{F}}/t}$ under certain conditions according to Lemma E.4. This is a key step in removing the $\sqrt{d_{\mathcal{F}}}$ gap in regret bound.

Recall that ADALEVEL ensures $w_t D_{t,l} \leq 2^l \alpha / \gamma$ and $w_t \bar{\sigma}_t \leq 2^l \alpha$ if $t \in \Psi_{T+1,l}$. This implies the confidence radius $\beta_{t,l} = \widetilde{O}(2^l \alpha \sqrt{\log N_{\mathcal{F}}})$.

---

**Algorithm 3** ML-VTR

---

**Require:** $\alpha, \gamma, L = \lceil \log_2 \frac{1}{\alpha} \rceil$, confidence radius $\{\beta_{k,l}\}_{k \geq 1, l \in [L]}$

1: $\widehat{f}_{1,l} \leftarrow 0$ for $l \in [L]$

2: $\Psi_{1,l}, \Psi_{1,1,l} \leftarrow \varnothing$ for $l \in \overline{[L]}, \widetilde{\Psi}_{1,\ell}, \widetilde{\Psi}_{1,1,\ell} \leftarrow \varnothing$ for $\ell \in \overline{[L]}$

3: **for** $k = 1, \ldots, K$ **do**

4: $\quad V_{k,H+1} \leftarrow 0$

5: $\quad$ **for** $h = H, \ldots, 1$ **do**

6: $\qquad Q_{k,h}(\cdot, \cdot) \leftarrow \min_{l \in [L]} \{1, r_h(\cdot, \cdot) + \widehat{f}_{k,l}(\cdot, \cdot, V_{k,h+1}) + \min\{1, \beta_{k,l} D_{k,l}(\cdot, \cdot, V_{k,h+1})\}\}$

7: $\qquad V_{k,h} \leftarrow \max_{a \in \mathcal{A}} Q_{k,h}(\cdot, a)$

8: $\qquad \pi_h^k \leftarrow \arg\max_{a \in \mathcal{A}} Q_{k,h}(\cdot, a)$

9: $\quad$ **end for**

10: $\quad$ Receive $s_1^k$

11: $\quad$ **for** $h = 1, \ldots, H$ **do**

12: $\qquad$ Take action $a_h^k \leftarrow \pi_h^k(s_h^k)$, receive $s_{h+1}^k$

13: $\qquad$ Update $z_{k,h} \leftarrow (s_h^k, a_h^k, V_{k,h+1}), \widetilde{z}_{k,h} \leftarrow (s_h^k, a_h^k, V_{k,h+1}^2), y_{k,h} \leftarrow V_{k,h+1}(s_{h+1}^k)$

14: $\qquad$ Update $\bar{\sigma}_{k,h}^2$ according to (5.2)

15: $\qquad$ Update $l_{k,h}, w_{k,h} \leftarrow \text{ADALEVEL}(\{D_{k,h,l}(z_{k,h})\}_{l \in [L]}, \bar{\sigma}_{k,h}, \alpha, \gamma)$

16: $\qquad$ Update $\ell_{k,h}, \widetilde{w}_{k,h} \leftarrow \text{ADALEVEL}(\{D_{k,h,l}(\widetilde{z}_{k,h})\}_{l \in [L]}, c_v \bar{\sigma}_{k,h}, \alpha, \gamma)$

17: $\qquad$ Update $\Psi_{k,h+1,l_{k,h}} \leftarrow \Psi_{k,h,l_{k,h}} \cup \{(k, h)\}, \Psi_{k,h+1,l} \leftarrow \Psi_{k,h,l}$ for $l \in \overline{[L]}, l \neq l_{k,h}$

18: $\qquad$ Update $\widetilde{\Psi}_{k,h+1,\ell_{k,h}} \leftarrow \widetilde{\Psi}_{k,h,\ell_{k,h}} \cup \{(k, h)\}, \widetilde{\Psi}_{k,h+1,\ell} \leftarrow \widetilde{\Psi}_{k,h,\ell}$ for $\ell \in \overline{[L]}, \ell \neq \ell_{k,h}$

19: $\quad$ **end for**

20: $\quad$ Update $\Psi_{k+1,l}, \Psi_{k+1,1,l} \leftarrow \Psi_{k,H+1,l}$ for $l \in [L], \widetilde{\Psi}_{k+1,\ell}, \widetilde{\Psi}_{k+1,1,\ell} \leftarrow \widetilde{\Psi}_{k,H+1,\ell}$ for $\ell \in [L]$

21: $\quad$ Update $\widehat{f}_{k+1,l}$ according to (5.1) for $l \in [L]$

22: **end for**

---

**Summation of Bonuses in Each Level** The regret can be related to the summation of bonuses across each level, as follows:

$$\text{Regret}(T) \leq 2 \sum_{l \in \overline{[L]}} \sum_{t \in \Psi_{T+1,l}} \min_{l \in [L]} \{R, \beta_{t,l} D_{t,l}(x_t)\}.$$

Thanks to ADALEVEL, the properties in Property 1 hold. Specifically, for any $l \in [L-1]$, if $t \in \Psi_{T+1,l}$, the maximum over uncertainty $D_{t,l}(x_t)$ and estimated variance $\bar{\sigma}_t$ is of the order $2^l$. For high-uncertainty data, $\beta_{t,l+1} D_{t,l+1}(x_t) \approx 2^{2l}$, while Lemma E.4 implies $|\Psi_{T+1,l}| \approx 2^{-2l} d_{\mathcal{F}}$, which leads to a lower order term in the final regret. For high-variance data, $\beta_{t,l} D_{t,l}(x_t) \approx \bar{\sigma}_t D_{t,l}(x_t)$, and Lemma E.4 implies $D_{t,l}(x_t) \approx \sqrt{d_{\mathcal{F}}/|\Psi_{T+1,l}|}$, resulting a second-order term in the final regret. We provide a more fine-grained analysis in Lemma E.3 to prove that for any $l \in [L-1]$,

$$\sum_{t \in \Psi_{T+1,l}} \min_{l \in [L]} \{R, \beta_{t,l} D_{t,l}(x_t)\} = \widetilde{O}\Big(\sqrt{d_{\mathcal{F}} \log N_{\mathcal{F}} \sum_{t \in \Psi_{T+1,l}} \bar{\sigma}_t^2} + R d_{\mathcal{F}} \log N_{\mathcal{F}}\Big).$$

The complete proof requires an in-depth analysis of the summation over different levels, and a careful treatment of estimated variance to eliminate lower-order terms.

## 5 MULTI-LEVEL REGRESSION FOR REINFORCEMENT LEARNING

In this section, we extend our multi-level regression framework to MDPs with general function approximation. This yields a new algorithm, ML-VTR, as detailed in Algorithm 3. We denote $D_{k,l}(\cdot) := D_{\mathcal{F}}(\cdot; z_{\Psi_{k,l}} \cup \widetilde{z}_{\widetilde{\Psi}_{k,l}}, w_{\Psi_{k,l}} \cup \widetilde{w}_{\widetilde{\Psi}_{k,l}}), D_{k,h,l}(\cdot) := D_{\mathcal{F}}(\cdot; z_{\Psi_{k,h,l}} \cup \widetilde{z}_{\widetilde{\Psi}_{k,h,l}}, w_{\Psi_{k,h,l}} \cup \widetilde{w}_{\widetilde{\Psi}_{k,h,l}})$. We first outline the high-level idea, then analyze the computational complexity and regret bound.

### 5.1 ALGORITHM DESCRIPTION

ML-VTR features a novel combination of the Multi-Level regression framework in Section 4 and Value-Targeted Regression (VTR) developed in Ayoub et al. (2020). Specifically, similar to

UCB-MLR, in Line 15, we leverage ADALEVEL to partition data into sets $\{\Psi_{K+1,l}\}_{l \in [L]}$ based on their uncertainty $\{D_{k,h,l}(z_{k,h})\}_{l \in [L]}$ and estimated variance $\bar{\sigma}_{k,h}$ for data points $z_{k,h}$. A similar process is applied to create the sets $\{\widetilde{\Psi}_{K+1,l}\}_{l \in [L]}$ similarly for data points $\widetilde{z}_{k,h}$. Here, $z_{i,h}$ and $\widetilde{z}_{i,h}$ is defined in Line 13. Then we adopt Multi-Level VTR to estimate the model. Since all data share the same transition model $f_*$, we can estimate it in a combined manner to reduce error:

$$\widehat{f}_{k+1,l} = \underset{f \in \mathcal{F}}{\operatorname{argmin}} \sum_{(i,h) \in \Psi_{k+1,l}} w_{i,h}^2 (f(z_{i,h}) - y_{i,h})^2 + \sum_{(i,h) \in \widetilde{\Psi}_{k+1,l}} \widetilde{w}_{i,h}^2 (f(\widetilde{z}_{i,h}) - y_{i,h}^2)^2. \quad (5.1)$$

Once the estimate $\{\widehat{f}_{k,l}\}_{l \in [L]}$ are obtained, we construct the action value functions $\{Q_{k,h}\}_{h \in [H]}$ as in Line 6. And the upper bound of $\operatorname{Var}[y_{k,h}|z_{k,h}] = [\mathbb{V}V_{k,h+1}](s_h^k, a_h^k)$ is then set as

$$\bar{\sigma}_{k,h}^2 = \min_{l \in [L], \ell \in [L]} \left\{ 1, \widehat{f}_{k,\ell}(\widetilde{z}_{k,h}) - \widehat{f}_{k,l}^2(z_{k,h}) + \min\{1, 2\beta_{k,l} D_{k,l}(z_{k,h})\} + \min\{1, \beta_{k,\ell} D_{k,\ell}(\widetilde{z}_{k,h})\} \right\}. \quad (5.2)$$

**Computational Complexity**   We analyze the computational complexity of ML-VTR under the assumption that for any $(s, a, V) \in \mathcal{S} \times \mathcal{A} \times \mathcal{V}$, the function $f_{\mathbb{P}}(s, a, V) = \sum_{s' \in \mathcal{S}} \mathbb{P}(s'|s, a)V(s')$ can be evaluated in $\mathcal{O}$ time. Recall $\mathcal{R}$ represents the computational cost of the regression oracle. We consider the computation cost for a single step $(k, h)$. First, computing the action value function $Q_{k,h}$ in Line 6 for a given state-action pair $(s, a)$ requires $\widetilde{O}(L(\mathcal{O} + \mathcal{R}))$ time, since it involves evaluating the estimated function $\widehat{f}_{k,l}$ and computing the uncertainty $\mathcal{D}_{k,l}$ for $l \in [L]$. To take an action based on $\pi_h^k$, the algorithm needs to compute $Q_{k,h}$ for $|\mathcal{A}|$ actions. Next, the estimated variance $\sigma_{k,h}^2$ in (5.2) can be computed within $\widetilde{O}(L(\mathcal{O} + \mathcal{R}))$ time. And ADALEVEL takes $\widetilde{O}(L)$ time. Finally, it takes $L\mathcal{R}$ time to calculate $\widehat{f}_{k+1,l}$ in (5.1) for $l \in [L]$. Therefore, the total computational cost of ML-VTR is $\widetilde{O}(KH|\mathcal{A}|(\mathcal{O} + \mathcal{R}))$.

## 5.2 REGRET BOUND

**Theorem 5.1.** For MDP with general function approximation defined in Section 3, if the parameters in Algorithm 3 are set according to Section C, then with probability at least $1 - (L + 2)\delta$, ML-VTR achieves

$$\operatorname{Regret}(K) = \widetilde{O}\left(\sqrt{d_{\mathcal{F}} \log N_{\mathcal{F}} \operatorname{Var}_K^*} + \max\{1, c_v\} d_{\mathcal{F}} \log N_{\mathcal{F}}\right),$$

where $\operatorname{Var}_K^*$ is the total variance of the optimal value functions $\{V_h^*\}_{h \in [H]}$:

$$\operatorname{Var}_K^* = \sum_{k=1}^{K} \sum_{h=1}^{H} [\mathbb{V}V_{h+1}^*](s_h^k, a_h^k). \quad (5.3)$$

*Proof.* The proof combines the technique used in proving contextual bandits with a fine-grained analysis of the higher-order moments of value functions, which eliminates polynomial dependence on the horizon $H$. See Section C for a detailed proof. □

Our second-order result from Theorem 5.1 is also horizon-free, as its dependence on the horizon $H$ is up to logarithmic factors. As a special case, for a $d$-dimensional linear mixture MDP, we have $d_{\mathcal{F}}, \log N_{\mathcal{F}} = O(d)$ (Huang et al., 2024). Our bound therefore simplifies to $\widetilde{O}(d\sqrt{\operatorname{Var}_K^*} + d^2)$, which matches the state-of-the-art result by Zhao et al. (2023). This demonstrates that our novel algorithm design and fine-grained analysis effectively and sharply handle general RL problems.

## 6 CONCLUSION

This paper presents a novel multi-level regression framework, ADALEVEL, that resolves a key challenge in online learning by partitioning data based on both uncertainty and variance. Our UCB-MLR algorithm for nonlinear contextual bandits, is the first to achieve an optimal second-order regret bound with computational efficiency. We extend this framework to reinforcement learning with general function approximation, where our ML-VTR algorithm provides the first horizon-free, second-order, and efficient regret bound. This multi-level regression technique is of independent interest and applicable to a broad range of online decision-making problems.

ETHICS STATEMENT

The authors have read and adhere to the ICLR Code of Ethics.

REPRODUCIBILITY STATEMENT

For reproducibility, we have included all necessary details in the main body and appendix. The full proofs for our theoretical results are in appendix.

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

## THE USE OF LARGE LANGUAGE MODELS (LLMS)

LLM was used as a general-purpose writing assistant for tasks like grammar and spelling correction, and to refine sentence structure. All authors take full responsibility for the final content.

## A  PROOF OF LEMMA 4.3

*Proof of Lemma 4.3.* We define filtration $\{\mathcal{G}_t\}_{t\geq 1}$ such that $X_t \in \mathcal{G}_{t-1}$, $Y_t \in \mathcal{G}_t$. Recall the definition of $\widehat{f}_{t+1}$ in (4.4), which implies

$$\sum_{s=1}^{t} w_s^2(\widehat{f}_{t+1}(X_s) - f_*(X_s))^2 \leq 2\sum_{s=1}^{t} w_s^2(\widehat{f}_{t+1}(X_s) - f_*(X_s))(Y_s - f_*(X_s)).$$

For any fixed $f \in \mathcal{F}$, denote $E_s(f) = w_s^2(f(X_s) - f_*(X_s))(Y_s - f_*(X_s))$, which is a martingale difference sequence adapted to the filtration $\{\mathcal{G}_s\}_{s\in[t]}$. Note $|w_s| \leq W$ and $f(X_s), f_*(X_s), Y_s$ are bounded in $[0, R]$, thereby the expectation and summation of variances are upper bounded by

$$|E_s(f)| \leq W^2 R^2, \quad \sum_{s=1}^{t} \mathbb{E}[E_s^2(f)|\mathcal{G}_{s-1}] \leq \sigma^2 W^2 R^2 t.$$

We denote $D_s = D_{\mathcal{F}}(X_s; X_{[s]}, w_{[s]})$ for short. Furthermore, we have

$$\max_{s\in[t]} |E_s(f)| \overset{(a)}{\leq} R\max_{s\in[t]} w_s^2 D_s \sqrt{\sum_{s=1}^{t} w_s^2(f(X_s) - f_*(X_s))^2 + \lambda},$$

$$\sum_{s=1}^{t} \mathbb{E}[E_s^2(f)|\mathcal{G}_{s-1}] \leq \sigma^2 \sum_{s=1}^{t} w_s^2(f(X_s) - f_*(X_s))^2,$$

where $(a)$ holds due to the definition of $D_{\mathcal{F}}$ in (3.1). Let $\epsilon > 0$ and $\mathcal{V}$ be a $\epsilon$-covering net of $\mathcal{F}$. Applying Lemma E.1 with $m = v = \sigma^2$, $\iota_t = 16\log\frac{2N_{\mathcal{F}}(\epsilon)t^2(\log(\sigma^2 W^2 R^2 t)+2)(\log(W^2 R^2)+2)}{\delta}$, and a union bound over $f \in \mathcal{V}$, for any $t \geq 1$, with probability at least $1 - \delta/(2t^2)$, we have for all $f \in \mathcal{V}$,

$$2\sum_{s=1}^{t} E_s(f)$$

$$\leq \sqrt{\iota_t} \cdot \sqrt{\sigma^2 \sum_{s=1}^{t} w_s^2(f(X_s) - f_*(X_s))^2 + \sigma^4}$$

$$\quad + \iota_t \left(L\max_{s\in[t]} w_s^2 D_s \cdot \sqrt{\sum_{s=1}^{t} w_s^2(f(X_s) - f_*(X_s))^2 + \lambda} + \sigma^2\right)$$

$$\overset{(a)}{\leq} \left(\sqrt{\iota_t}\sigma + \iota_t L\max_{s\in[t]} w_s^2 D_s\right)\sqrt{\sum_{s=1}^{t} w_s^2(f(X_s) - f_*(X_s))^2} + \sqrt{\lambda}\iota_t L\max_{s\in[t]} w_s^2 D_s + 2\iota_t\sigma^2$$

$$\overset{(b)}{\leq} \frac{1}{2}\sum_{s=1}^{t} w_s^2(f(X_s) - f_*(X_s))^2 + \frac{1}{2}\left(\sqrt{\iota_t}\sigma + \iota_t L\max_{s\in[t]} w_s^2 D_s\right)^2 + \frac{1}{2}\left(\iota_t L\max_{s\in[t]} w_s^2 D_s\right)^2$$

$$\quad + \frac{1}{2}\lambda + 2\iota_t\sigma^2$$

$$\overset{(c)}{\leq} \frac{1}{2}\sum_{s=1}^{t} w_s^2(f(X_s) - f_*(X_s))^2 + \frac{1}{2}\left(\sqrt{\iota_t}\sigma + 2\iota_t L\max_{s\in[t]} w_s^2 D_s\right)^2 + \frac{1}{2}\lambda + 2\iota_t\sigma^2,$$

where $(a)$ holds due to $\sqrt{a+b} \leq \sqrt{a} + \sqrt{b}$ for any $a, b \geq 0$ and $\iota_t \geq 1$, $(b)$ holds due to $\sqrt{ab} \leq a/2 + b/2$ for any $a, b \geq 0$, and $(c)$ holds due to $a^2 + b^2 \leq (a+b)^2$ for any $a, b \geq 0$. Let $g \in \mathcal{V}$

such that $\|g - \widehat{f}_{t+1}\|_\infty \leq \epsilon$, then

$$\sum_{s=1}^{t} w_s^2(\widehat{f}_{t+1}(X_s) - f_*(X_s))^2$$

$$\leq 2\sum_{s=1}^{t} w_s^2(\widehat{f}_{t+1}(X_s) - f_*(X_s))(Y_s - f_*(X_s))$$

$$\leq 2\sum_{s=1}^{t} w_s^2(g(X_s) - f_*(X_s))(Y_s - f_*(X_s)) + 2W^2 Rt\epsilon$$

$$\leq \frac{1}{2}\sum_{s=1}^{t} w_s^2(g(X_s) - f_*(X_s))^2 + \frac{1}{2}\left(\sqrt{\iota_t}\sigma + 2\iota_t R \max_{s\in[t]} w_s^2 D_s\right)^2 + \frac{1}{2}\lambda + 2\iota_t\sigma^2 + 2W^2 Rt\epsilon$$

$$\leq \frac{1}{2}\sum_{s=1}^{t} w_s^2(\widehat{f}_{t+1}(X_s) - f_*(X_s))^2 + \frac{1}{2}\left(\sqrt{\iota_t}\sigma + 2\iota_t R \max_{s\in[t]} w_s^2 D_s\right)^2 + \frac{1}{2}\lambda + 2\iota_t\sigma^2 + 3W^2 Rt\epsilon.$$

That is for any fixed $t > 0$, we have

$$\sum_{s=1}^{t} w_s^2(\widehat{f}_{t+1}(X_s) - f_*(X_s))^2 \leq \left(\sqrt{\iota_t}\sigma + 2\iota_t R \max_{s\in[t]} w_s^2 D_s\right)^2 + \lambda + 4\iota_t\sigma^2 + 6W^2 Rt\epsilon$$

$$\leq \left(3\sqrt{\iota_t}\sigma + 2\iota_t R \max_{s\in[t]} w_s^2 D_s + \sqrt{\lambda} + \sqrt{6W^2 Rt\epsilon}\right)^2,$$

where the second inequality is due to $2\sqrt{ab} \leq a + b$ and $a + b \leq (\sqrt{a} + \sqrt{b})^2$ for any $a, b \geq 0$. Note

$$w_t^2 D_t = w_t^2 D_\mathcal{F}(X_t; X_{[t]}, w_{[t]})$$

$$= \sup_{f_1, f_2 \in \mathcal{F}} \frac{w_t^2(f_1(X_t) - f_2(X_t))^2}{\sum_{s\in[t]} w_s^2(f_1(X_s) - f_2(X_s))^2 + \lambda}$$

$$\leq \min\{1, w_t^2 D_\mathcal{F}(X_t; X_{[t-1]}, w_{[t-1]})\}.$$

Finally, the result holds through a union bound over all $t \geq 1$ and $\sum_{t=1}^{\infty} \frac{1}{2t^2} \leq 1$. $\qquad\square$

## B  PROOF OF THEOREM 4.2

**Parameters in Algorithm 1**  For any $t \in [T]$, $l \in [L]$, $\ell \in [\widetilde{L}]$, let $\mathcal{B}_{t,l}$, $\widetilde{\mathcal{B}}_{t,\ell}$ denote the confidence region as follows:

$$\mathcal{B}_{t,l} := \left\{f \in \mathcal{F}: \sum_{s\in\Psi_{t,l}} w_s^2(\widehat{f}_{t,l}(x_s) - f(x_s)^2 \leq \beta_{t,l}^2\right\},$$

$$\widetilde{\mathcal{B}}_{t,\ell} := \left\{g \in \mathcal{G}: \sum_{s\in\widetilde{\Psi}_{t,\ell}} \widetilde{w}_s^2(\widehat{g}_{t,\ell}(x_s) - g(x_s)^2 \leq \widetilde{\beta}_{t,\ell}^2\right\}.$$

Here

$$\beta_{t,l} = 2^l\alpha\left(3\sqrt{\iota_t} + 2\frac{\iota_t R}{\gamma}\right) + \sqrt{\lambda} + \sqrt{6Rt\epsilon}, \qquad (\text{B.1})$$

$$\widetilde{\beta}_{t,\ell} = 2^\ell\widetilde{\alpha}\left(3\sqrt{\widetilde{\iota}_t} + 2\frac{\widetilde{\iota}_t R^2}{\widetilde{\gamma}}\right) + \sqrt{\widetilde{\lambda}} + \sqrt{6R^2 t\epsilon}, \qquad (\text{B.2})$$

where

$$\iota_t = 16\log\frac{2N_\mathcal{F}(\epsilon)t^2(\log(\sigma^2 R^2 t) + 2)(\log(R^2) + 2)}{\delta} = \widetilde{O}(\log N_\mathcal{F})$$

$$\widetilde{\iota}_t = 16\log\frac{2N_\mathcal{G}(\widetilde{\epsilon})t^2(\log(c_v^2\sigma^2 R^4 t) + 2)(\log(R^4) + 2)}{\delta} = \widetilde{O}(\log N_\mathcal{G}).$$

Furthermore, setting

$$\gamma = R\sqrt{\log N_{\mathcal{F}}}, \quad \widetilde{\gamma} = R^2\sqrt{\log N_{\mathcal{G}}}, \tag{B.3}$$

$$\lambda = \alpha^2 \log N_{\mathcal{F}}, \quad \widetilde{\lambda} = \widetilde{\alpha}^2 \log N_{\mathcal{G}}, \tag{B.4}$$

$$\epsilon = \frac{\alpha^2 \log N_{\mathcal{F}}}{RT}, \quad \widetilde{\epsilon} = \frac{\widetilde{\alpha}^2 \log N_{\mathcal{G}}}{R^2 T}, \tag{B.5}$$

we have

$$\beta_{t,l} = \widetilde{O}(2^l \alpha \sqrt{\log N_{\mathcal{F}}}), \quad \widetilde{\beta}_{t,\ell} = \widetilde{O}(2^\ell \widetilde{\alpha} \sqrt{\log N_{\mathcal{G}}}).$$

**Property 1** (Properties of ADALEVEL). For any $t \in [T]$, suppose $l_t = l$, then

1. If $l = 0$:
$$D_{t,1}(X_t) \leq 2\alpha/\gamma.$$

2. If $l \in [L-1]$:
$$\begin{cases} w_t = \frac{2^l \alpha}{\gamma D_{t,l}(X_t)}, \\ D_{t,l+1}(X_t) \leq 2^{l+1}\alpha/\gamma, \\ \bar{\sigma}_t \leq 2^l \alpha; \end{cases} \quad \text{or} \quad \begin{cases} w_t = 1, \\ D_{t,l}(X_t) \leq 2^l \alpha/\gamma, \\ 2^{l-1}\alpha < \bar{\sigma}_t \leq 2^l \alpha. \end{cases}$$

3. If $l = L$:
$$\begin{cases} w_t = \frac{2^L \alpha}{\gamma D_{t,L}(X_t)}, \\ \bar{\sigma}_t \leq 2^L \alpha; \end{cases} \quad \text{or} \quad \begin{cases} w_t = 1, \\ D_{t,L}(X_t) \leq 2^L \alpha/\gamma, \\ 2^{L-1}\alpha < \bar{\sigma}_t \leq 2^L \alpha. \end{cases}$$

*Proof of Theorem 4.2.* For $t \in [T]$, we define events $\mathcal{E}_t, \mathcal{E}$ as

$$\mathcal{E}_t = \{\forall l \in [L], f_* \in \mathcal{B}_{t,l} \text{ and } \forall \ell \in [\widetilde{L}], g_* \in \widetilde{\mathcal{B}}_{t,\ell}\}, \quad \mathcal{E} = \bigcap_{k \in [K]} \mathcal{E}_t.$$

The following lemmas hold.

**Lemma B.1.** On event $\mathcal{E}_t$, we have for all $l \in [L], \ell \in [\widetilde{L}]$,

$$|\widehat{f}_{t,l}(x) - f_*(x)| \leq \beta_{t,l} D_{t,l}(x),$$

$$|\widehat{g}_{t,\ell}(x) - g_*(x)| \leq \widetilde{\beta}_{t,\ell}\widetilde{D}_{t,\ell}(x).$$

Furthermore,

$$f_*(x) \leq \min_{l \in [L]} \left(\widehat{f}_{t,l}(x) + \min\{R, \beta_{t,l} D_{t,l}(x)\}\right),$$

and

$$\sigma_t^2 \leq \bar{\sigma}_t^2.$$

*Proof.* On event $\mathcal{E}_t$, for any $l \in [L]$, we have

$$|\widehat{f}_{t,l}(x) - f_*(x)| \leq D_{t,l}(x)\sqrt{\sum_{s \in \Psi_{t,l}} w_s^2(\widehat{f}_{t,l}(x_s) - f_*(x))^2 + \lambda}$$

$$\leq D_{t,l}(x)\sqrt{\beta_{t,l}^2 + \lambda}$$

$$\approx \beta_{t,l} D_{t,l}(x),$$

since $\sqrt{\lambda} = O(\beta_{t,l})$ according to (B.4). Similarly, for any $\ell \in [L]$,

$$|\widehat{g}_{t,\ell}(x) - g_*(x)| \leq \widetilde{\beta}_{t,l}\widetilde{D}_{t,l}(x).$$

Furthermore, since this holds for all $l \in [L]$, we can choose the upper confidence bound of $f_*(x)$ as

$$\min_{l \in [L]} \left(\widehat{f}_{t,l}(x) + \min\{R, \beta_{t,l} D_{t,l}(x)\}\right) \geq f_*(x).$$

Recall the definition of $\bar{\sigma}_t$ in (4.3), we have for all $l \in [L], \ell \in [\widetilde{L}]$,

$$|(\widehat{g}_{t,\ell}(x_t) - \widehat{f}_{t,l}^2(x_t)) - (g_*(x_t) - f_*^2(x_t))|$$

$$\leq |\widehat{g}_{t,\ell}(x_t) - g_*(x_t)| + |\widehat{f}_{t,l}(x_t) + f_*(x_t)| \cdot |\widehat{f}_{t,l}(x_t) - f_*(x_t))|$$

$$\leq \min\{R^2, \widetilde{\beta}_{t,\ell} \widetilde{D}_{t,\ell}(x_t)\} + \min\{R^2, 2R\beta_{t,l} D_{t,l}(x_t)\}.$$

Therefore, $\sigma_t^2$ is bounded by $\bar{\sigma}_t^2$:

$$\sigma_t^2 = \mathrm{Var}[y_t^2 | x_t] = \mathbb{E}[y_t^2 | x_t] - \mathbb{E}^2[y_t | x_t] = g_*(x_t) - f_*^2(x_t)$$

$$\leq \min_{l \in [L], \ell \in [\widetilde{L}]} \left\{\sigma^2, \widehat{g}_{t,\ell}(x_t) - \widehat{f}_{t,l}^2(x_t) + R\min\{R, 2\beta_{t,l} D_{t,l}(x_t)\} + \min\{R^2, \widetilde{\beta}_{t,\ell} \widetilde{D}_{t,\ell}(x_t)\}\right\}$$

$$= \bar{\sigma}_t^2.$$

$\square$

**Lemma B.2.** Event $\mathcal{E}$ holds with probability at least $1 - (L + \widetilde{L})\delta$.

*Proof.* By a union bound, with probability at least $1 - (L + \widetilde{L})\delta$, the result follows from Lemma 4.3 using $\{X_t, Y_t, w_t\}_t = \{x_t, y_t, w_t\}_{t \in \Psi_{T+1,l}}$, $\mathcal{F}$ for $l \in [L]$, and using $\{X_t, Y_t, w_t\}_t = \{x_t, y_t^2, \widetilde{w}_t\}_{t \in \widetilde{\Psi}_{T+1,\ell}}$, $\mathcal{F} = \mathcal{G}$ for $\ell \in [\widetilde{L}]$. We will check the conditions of Lemma 4.3 for all $t \in [T]$ by induction.

First, for $t = 1$, the result holds trivially.

Next, for $t > 1$, suppose event $\bigcap_{s \in [t]} \mathcal{E}_s$ holds, by Lemma B.1, we have for all $s \in [t]$,

$$\sigma_s^2 \leq \bar{\sigma}_s^2.$$

Thus from Property 1, for all $l \in [L], \ell \in [L]$,

$$\mathrm{Var}[y_s | x_s] = \sigma_s^2 \leq \bar{\sigma}_s^2 \leq 2^l \alpha, \quad w_s D_{s,l}(x_s) \leq \frac{2^l \alpha}{\gamma}, \quad \forall s \in \Psi_{t+1,l},$$

$$\mathrm{Var}[y_s^2 | x_s] \leq c_v^2 \sigma_s^2 \leq c_v^2 \bar{\sigma}_s^2 \leq 2^\ell \widetilde{\alpha}, \quad \widetilde{w}_s \widetilde{D}_{s,\ell}(x_s) \leq \frac{2^\ell \widetilde{\alpha}}{\widetilde{\gamma}}, \quad \forall s \in \widetilde{\Psi}_{t+1,\ell}.$$

Applying Lemma 4.3 with $\sigma = 2^l \alpha$, $\max_{s \in [t]} w_s^2 D_{\mathcal{F}}(X_s; X_{[s-1]}, w_{[s-1]}) = \frac{2^l \alpha}{\gamma}$, we have

$$\sum_{s \in \Psi_{t+1,l}} w_s^2 (\widehat{f}_{t+1,l}(x_s) - f_*(x_s))^2 \leq \beta_{t+1,l}^2,$$

that is $f_* \in \mathcal{B}_{t+1,l}$ for all $l \in [L]$. Applying Lemma 4.3 again with $\sigma = 2^\ell \widetilde{\alpha}$, $\max_{s \in [t]} w_s^2 D_{\mathcal{F}}(X_s; X_{[s-1]}, w_{[s-1]}) = \frac{2^\ell \widetilde{\alpha}}{\widetilde{\gamma}}$, we have

$$\sum_{s \in \widetilde{\Psi}_{t+1,\ell}} \widetilde{w}_s^2 (\widehat{g}_{t+1,\ell}(x_s) - g_*(x_s))^2 \leq \widetilde{\beta}_{t+1,\ell}^2,$$

that is $g_* \in \widetilde{\mathcal{B}}_{t+1,\ell}$ for all $\ell \in [\widetilde{L}]$, so event $\mathcal{E}_{t+1}$ holds.

Then the proof is completed by induction over $t \in [T]$. $\square$

We define

$$U := \sum_{t \in [T]} \min_{l \in [L]} \{R, \beta_{t,l} D_{t,l}(x_t)\}, \tag{B.6}$$

$$\widetilde{U} := \sum_{t \in [T]} \min_{\ell \in [\widetilde{L}]} \{R^2, \widetilde{\beta}_{t,\ell} \widetilde{D}_{t,\ell}(x_t)\}. \tag{B.7}$$

On event $\mathcal{E}$, which holds with probability at least $1 - (L + \widetilde{L})\delta$ by Lemma B.2, by the optimism of $x_t$ implied by Lemma B.1, regret is bounded as

$$
\begin{aligned}
\text{Regret}(T) &= \sum_{t \in [T]} (f_*(x_t^*) - f_*(x_t)) \\
&\leq \sum_{t \in [T]} \Big[ \min_{l \in [L]} \big(\widehat{f}_{t,l}(x_t) + \min\{R, \beta_{t,l} D_{t,l}(x_t)\}\big) - f_*(x_t) \Big] \\
&\leq 2 \sum_{t \in [T]} \min_{l \in [L]} \{R, \beta_{t,l} D_{t,l}(x_t)\} \\
&\overset{\text{(B.6)}}{=} 2U.
\end{aligned}
$$
(B.8)

Setting

$$
\alpha = R\sqrt{\frac{d_{\mathcal{F}} \log N_{\mathcal{F}}}{T}}, \quad \widetilde{\alpha} = R^2\sqrt{\frac{d_{\mathcal{G}} \log N_{\mathcal{G}}}{T}},
$$
(B.9)

applying Lemma E.3 to $U$ and $\widetilde{U}$, we have

$$
U = \widetilde{O}\Big(\sqrt{d_{\mathcal{F}} \log N_{\mathcal{F}} \sum\nolimits_{t \in [T]} \bar{\sigma}_t^2} + R d_{\mathcal{F}} \log N_{\mathcal{F}}\Big),
$$
(B.10)

$$
\widetilde{U} = \widetilde{O}\Big(c_v R\sqrt{d_{\mathcal{F}} \log N_{\mathcal{F}} \sum\nolimits_{t \in [T]} \bar{\sigma}_t^2} + R^2 d_{\mathcal{G}} \log N_{\mathcal{G}}\Big).
$$
(B.11)

Furthermore,

$$
\begin{aligned}
\sum_{t \in [T]} \bar{\sigma}_t^2 &\leq \sum_{t \in [T]} \big(\sigma_t^2 + 2R \min_{l \in [L]}\{R, 2\beta_{t,l} D_{t,l}(x_t)\} + 2 \min_{\ell \in [\widetilde{L}]}\{R^2, \widetilde{\beta}_{t,\ell} \widetilde{D}_{t,\ell}(x_t)\}\big) \\
&\leq \sum_{t \in [T]} \sigma_t^2 + 4RU + 2\widetilde{U}.
\end{aligned}
$$
(B.12)

So we have

$$
\begin{aligned}
\widetilde{U} &\overset{\text{(B.11)}}{\lesssim} c_v R\sqrt{d_{\mathcal{G}} \log N_{\mathcal{G}} \sum\nolimits_{t \in [T]} \bar{\sigma}_t^2} + R^2 d_{\mathcal{G}} \log N_{\mathcal{G}} \\
&\overset{\text{(B.12)}}{\lesssim} c_v R\sqrt{d_{\mathcal{G}} \log N_{\mathcal{G}}\Big(\sum\nolimits_{t \in [T]} \sigma_t^2 + RU + \widetilde{U}\Big)} + R^2 d_{\mathcal{G}} \log N_{\mathcal{G}} \\
&\lesssim c_v R\sqrt{d_{\mathcal{G}} \log N_{\mathcal{G}}\Big(\sum\nolimits_{t \in [T]} \sigma_t^2 + RU\Big)} + \max\{1, c_v^2\} R^2 d_{\mathcal{G}} \log N_{\mathcal{G}},
\end{aligned}
$$
(B.13)

where the last inequality holds since $x \leq a\sqrt{x} + b$ implies $x \leq a^2 + 2b$ for any $x \geq 0$.

And

$$
\begin{aligned}
U &\overset{\text{(B.10)}}{\lesssim} \sqrt{d_{\mathcal{F}} \log N_{\mathcal{F}} \sum\nolimits_{t \in [T]} \bar{\sigma}_t^2} + R d_{\mathcal{F}} \log N_{\mathcal{F}} \\
&\overset{\text{(B.12)}}{\lesssim} \sqrt{d_{\mathcal{F}} \log N_{\mathcal{F}}\Big(\sum\nolimits_{t \in [T]} \sigma_t^2 + RU + \widetilde{U}\Big)} + R d_{\mathcal{F}} \log N_{\mathcal{F}} \\
&\lesssim \sqrt{d_{\mathcal{F}} \log N_{\mathcal{F}}\Big(\sum\nolimits_{t \in [T]} \sigma_t^2 + RU\Big)} + R d_{\mathcal{F}} \log N_{\mathcal{F}} + \sqrt{d_{\mathcal{F}} \log N_{\mathcal{F}} \widetilde{U}},
\end{aligned}
$$

where the last inequality holds due to $\sqrt{a+b} \leq \sqrt{a} + \sqrt{b}$ for any $a, b \geq 0$. Here

$$\sqrt{d_{\mathcal{F}} \log N_{\mathcal{F}} \widetilde{U}}$$

$$= \sqrt{\max\{1, c_v\} R \sqrt{d_{\mathcal{F}} \log N_{\mathcal{F}} d_{\mathcal{G}} \log N_{\mathcal{G}}} \cdot \frac{\sqrt{\frac{d_{\mathcal{F}} \log N_{\mathcal{F}}}{d_{\mathcal{G}} \log N_{\mathcal{G}}}}}{\max\{1, c_v\} R} \widetilde{U}}$$

$$\lesssim \sqrt{d_{\mathcal{F}} \log N_{\mathcal{F}} \Big( \sum_{t \in [T]} \sigma_t^2 + RU \Big)} + \max\{1, c_v\} R \sqrt{d_{\mathcal{F}} \log N_{\mathcal{F}} d_{\mathcal{G}} \log N_{\mathcal{G}}}$$

$$\approx \sqrt{d_{\mathcal{F}} \log N_{\mathcal{F}} \Big( \sum_{t \in [T]} \sigma_t^2 + RU \Big)} + CR d_{\mathcal{F}} \log N_{\mathcal{F}},$$

where the inequality holds due to Cauchy-Schwartz inequality and (B.13), and the last equality holds since $C := \max\{1, c_v\} \sqrt{\frac{d_{\mathcal{G}} \log N_{\mathcal{G}}}{d_{\mathcal{F}} \log N_{\mathcal{F}}}}$. Plugin back, we have

$$U \lesssim \sqrt{d_{\mathcal{F}} \log N_{\mathcal{F}} \Big( \sum_{t \in [T]} \sigma_t^2 + RU \Big)} + \max\{1, C\} R d_{\mathcal{F}} \log N_{\mathcal{F}}$$

$$\lesssim \sqrt{d_{\mathcal{F}} \log N_{\mathcal{F}} \sum_{t \in [T]} \sigma_t^2} + \max\{1, C\} R d_{\mathcal{F}} \log N_{\mathcal{F}}. \tag{B.14}$$

Finally, combining (B.8) and (B.14), we have

$$\text{Regret}(T) = \widetilde{O}\Big( \sqrt{d_{\mathcal{F}} \log N_{\mathcal{F}} \sum_{t \in [T]} \sigma_t^2} + \max\{1, C\} R d_{\mathcal{F}} \log N_{\mathcal{F}} \Big).$$

$\square$

## C  PROOFS FOR REINFORCEMENT LEARNING

**Parameters in Algorithm 3**  For $k \in [K], l \in [L]$, let $\mathcal{B}_{k,l}$ denote the confidence region as follows:

$$\mathcal{B}_{k,l} := \Big\{ f \in \mathcal{F} : \sum_{(i,h) \in \Psi_{k,l}} w_{i,h}^2 (\widehat{f}_{k,l}(z_{i,h}) - f(z_{i,h}))^2 + \sum_{(i,h) \in \widetilde{\Psi}_{k,l}} \widetilde{w}_{i,h}^2 (\widehat{f}_{k,l}(\widetilde{z}_{i,h}) - f(\widetilde{z}_{i,h}))^2 \leq \beta_{k,l}^2 \Big\}.$$

Here

$$\beta_{k,l} = 2^l \alpha \Big( 3\sqrt{\iota_k} + 2\frac{\iota_k}{\gamma} \Big) + \sqrt{\lambda} + \sqrt{12kH\epsilon} \tag{C.1}$$

where

$$\iota_k = 16 \log \frac{16 N_{\mathcal{F}}(\epsilon) k^2 H^2 (\log(2kH) + 2)}{\delta} = \widetilde{O}(\log N_{\mathcal{F}}).$$

Furthermore, setting

$$\gamma = \sqrt{\log N_{\mathcal{F}}}, \tag{C.2}$$

$$\lambda = \alpha^2 \log N_{\mathcal{F}}, \tag{C.3}$$

$$\epsilon = \frac{\alpha^2 \log N_{\mathcal{F}}}{KH}, \tag{C.4}$$

we have

$$\beta_{k,l} = \widetilde{O}(2^l \alpha \sqrt{\log N_{\mathcal{F}}}).$$

### C.1  OPTIMISM

For $k \in [K]$, we define events $\mathcal{E}_k$, and $\mathcal{E}$ as

$$\mathcal{E}_k = \{\forall l \in [L], f_* \in \mathcal{B}_{k,l}\}, \quad \mathcal{E} = \bigcap_{k \in [K]} \mathcal{E}_k.$$

The following lemmas hold.

**Lemma C.1.** On event $\mathcal{E}_k$, we have for all $l \in [L]$,

$$|\widehat{f}_{k,l}(z) - f_*(z)| \le \beta_{k,l} D_{k,l}(z).$$

Furthermore, for all $h \in [H]$,

$$r_h(s_h^k, a_h^k) + [\mathbb{P}V_{k,h+1}](s_h^k, a_h^k) \le V_{k,h}(s_h^k),$$

$$V_{k,h}(s_h^k) - r_h(s_h^k, a_h^k) - [\mathbb{P}V_{k,h+1}](s_h^k, a_h^k) \le 2\min_{l \in [L]}\{1, \beta_{k,l}D_{k,l}(z_{k,h})\},$$

and

$$[\mathbb{V}V_{k,h+1}](s_h^k, a_h^k) \le \bar{\sigma}_{k,h}^2,$$

$$\bar{\sigma}_{k,h}^2 - [\mathbb{V}V_{k,h+1}](s_h^k, a_h^k) \le 2\min_{l \in [L]}\{1, 2\beta_{k,l}D_{k,l}(z_{k,h})\} + 2\min_{\ell \in [L]}\{1, \beta_{k,\ell}D_{k,\ell}(\widetilde{z}_{k,h})\}.$$

*Proof.* See Appendix D.1 for a detailed proof. □

**Lemma C.2.** Event $\mathcal{E}$ holds with probability at least $1 - L\delta$.

*Proof.* See Appendix D.2 for a detailed proof. □

**Lemma C.3.** On event $\mathcal{E}$, we have for all $(k, h) \in [K] \times [H]$, $Q_{k,h}(\cdot, \cdot) \ge Q_h^*(\cdot, \cdot), V_{k,h}(\cdot) \ge V_h^*(\cdot)$.

*Proof.* See Appendix D.3 for a detailed proof. □

## C.2 HIGHER-ORDER QUANTITIES IN MDPS

Inspired by Zhao et al. (2023); Huang et al. (2024), we define the following higher-order quantities of MDPs. Let $M = \lceil \log_2(3KH) \rceil$.

We define $\mathcal{K}$ to be a set of episodes when the sum of uncertainty within each level grows smoothly:

$$\mathcal{K} := \{k \in [K] : \forall l \in [L], \sum_{h \in \Psi_{k+1,l} \setminus \Psi_{k,l}} w_{k,h}^2 D_{k,h,l}^2(z_{k,h}) + \sum_{h \in \widetilde{\Psi}_{k+1,l} \setminus \widetilde{\Psi}_{k,l}} \widetilde{w}_{k,h}^2 D_{k,h,l}^2(\widetilde{z}_{k,h}) \le 1\}. \tag{C.5}$$

Let $\widetilde{\mathcal{K}} := [K] \setminus \mathcal{K}$. We can prove the number of episodes when uncertainty grows sharply is at most $|\widetilde{\mathcal{K}}| = \widetilde{O}(Ld_{\mathcal{F}})$.

We use $\check{V}_{k,h}(s)$ to denote the estimation error between the estimated value function and the optimal value function, and use $\widetilde{V}_{k,h}(s)$ to denote the sub-optimality gap of policy $\pi^k$ at stage $h$:

$$\check{V}_{k,h}(s) = V_{k,h}(s) - V_h^*(s), \quad \forall s \in \mathcal{S}, (k, h) \in [K] \times [H], \tag{C.6}$$

$$\widetilde{V}_{k,h}(s) = V_h^*(s) - V_h^{\pi^k}(s), \quad \forall s \in \mathcal{S}, (k, h) \in [K] \times [H]. \tag{C.7}$$

In addition, we use $\check{S}_m, \widetilde{S}_m$ to represent the total variance of $2^m$-th order value functions $\check{V}_{k,h+1}^{2^m}$, $\widetilde{V}_{k,h+1}^{2^m}$:

$$\check{S}_m = \sum_{k \in \mathcal{K}} \sum_h [\mathbb{V}\check{V}_{k,h+1}^{2^m}](s_h^k, a_h^k), \tag{C.8}$$

$$\widetilde{S}_m = \sum_{k \in \mathcal{K}} \sum_h [\mathbb{V}\widetilde{V}_{k,h+1}^{2^m}](s_h^k, a_h^k). \tag{C.9}$$

Then, for $2^m$-th order value functions $\check{V}_{k,h+1}^{2^m}, \widetilde{V}_{k,h+1}^{2^m}$, we use $\check{A}_m, \widetilde{A}_m$ to denote the summation of stochastic transition noise as follows:

$$\check{A}_m = \left| \sum_{k \in \mathcal{K}} \sum_h \left[ [\mathbb{P}\check{V}_{k,h+1}^{2^m}](s_h^k, a_h^k) - \check{V}_{k,h+1}^{2^m}(s_{h+1}^k) \right] \right|, \tag{C.10}$$

$$\widetilde{A}_m = \left| \sum_{k \in \mathcal{K}} \sum_h \left[ [\mathbb{P}\widetilde{V}_{k,h+1}^{2^m}](s_h^k, a_h^k) - \widetilde{V}_{k,h+1}^{2^m}(s_{h+1}^k) \right] \right|. \tag{C.11}$$

Finally, we use the $R, \widetilde{R}$ to denote the summation of bonuses:

$$R = \sum_{k \in \mathcal{K}} \sum_h \min_{l \in [L]} \{1, \beta_{k,l} D_{k,l}(z_{k,h})\}, \tag{C.12}$$

$$\widetilde{R} = \sum_{k \in \mathcal{K}} \sum_h \min_{\ell \in [L]} \{1, \beta_{k,\ell} D_{k,\ell}(\widetilde{z}_{k,h})\} \tag{C.13}$$

Now, we introduce the following lemmas to build the connection between these quantities.

**Lemma C.4.** We have

$$|\widetilde{\mathcal{K}}| \leq 2L d_{\mathcal{F}}. \tag{C.14}$$

*Proof.* See Appendix D.4 for a detailed proof. $\square$

**Lemma C.5.** On event $\mathcal{E}$, we have for all $m \in \overline{[M]}$,

$$\check{S}_m \leq \check{A}_{m+1} + 2^{m+1} \cdot (2R), \tag{C.15}$$

$$\widetilde{S}_m \leq \widetilde{A}_{m+1} + 2^{m+1} \cdot (2R + \check{A}_0). \tag{C.16}$$

*Proof.* See Appendix D.5 for a detailed proof. $\square$

**Lemma C.6.** With probability at least $1 - 2\delta$, we have for all $m \in \overline{[M]}$,

$$\check{A}_m \leq \sqrt{\zeta \check{S}_m} + \zeta, \tag{C.17}$$

$$\widetilde{A}_m \leq \sqrt{\zeta \check{S}_m} + \zeta, \tag{C.18}$$

where $\zeta = 8 \log(2(M+1)(\log(KH) + 2)/\delta)$. We denote the corresponding event by $\mathcal{A}$.

*Proof.* See Appendix D.6 for a detailed proof. $\square$

**Lemma C.7.** On event $\mathcal{E} \cap \mathcal{A}$, we have

$$\check{A}_0 \leq 2\sqrt{2\zeta R} + 7\zeta, \tag{C.19}$$

$$\check{A}_1 \leq 4\sqrt{\zeta R} + 7\zeta. \tag{C.20}$$

*Proof.* See Appendix D.7 for a detailed proof. $\square$

**Lemma C.8.** On event $\mathcal{E} \cap \mathcal{A}$, we have

$$\widetilde{A}_0 \leq 4\sqrt{2\zeta R} + 15\zeta. \tag{C.21}$$

*Proof.* See Appendix D.8 for a detailed proof. $\square$

**Lemma C.9.** Setting

$$\alpha = \frac{d_{\mathcal{F}} \log N_{\mathcal{F}}}{KH}, \tag{C.22}$$

on event $\mathcal{E} \cap \mathcal{A}$, we have

$$R = \widetilde{O}\left(\sqrt{d_{\mathcal{F}} \log N_{\mathcal{F}} \operatorname{Var}_K^*} + \max\{1, c_v\} d_{\mathcal{F}} \log N_{\mathcal{F}}\right). \tag{C.23}$$

*Proof.* See Appendix D.9 for a detailed proof. $\square$

## C.3 REGRET ANALYSIS

*Proof of Theorem 5.1.* On event $\mathcal{E} \cap \mathcal{A}$, which holds with probability at least $1 - (L + 2)\delta$ by Lemma C.2, C.6 and a union bound, we have all lemmas in this section hold. By the optimism implied by Lemma C.3, we have

$$\text{Regret}(K) = \sum_{k=1}^{K} \left(V_1^*(s_1^k) - V_1^{\pi^k}(s_1^k)\right) \leq \sum_{k=1}^{K} \left(V_{k,1}(s_1^k) - V_1^{\pi^k}(s_1^k)\right). \quad (C.24)$$

We further use Lemma C.10 to bound the regret with the quantities defined in Section C.2.

**Lemma C.10.** On event $\mathcal{E}$, we have

$$\sum_{k=1}^{K} \left(V_{k,1}(s_1^k) - V^{\pi^k}(s_1^k)\right) \leq 2R + \check{A}_0 + \widetilde{A}_0 + |\widetilde{\mathcal{K}}|. \quad (C.25)$$

*Proof.* First, we decompose $V_{k,1}(s_1^k)$ and $V_1^{\pi^k}(s_1^k)$ as follows

$$V_{k,1}(s_1^k) = \sum_{h=1}^{H}[V_{k,h}(s_h^k) - V_{k,h+1}(s_{h+1}^k)]$$

$$= \sum_{h=1}^{H} r(s_h^k, a_h^k) + \sum_{h=1}^{H} \left[V_{k,h}(s_h^k) - r(s_h^k, a_h^k) - [\mathbb{P}V_{k,h+1}](s_h^k, a_h^k)\right]$$

$$+ \sum_{h=1}^{H} \left[[\mathbb{P}V_{k,h+1}](s_h^k, a_h^k) - V_{k,h+1}(s_{h+1}^k)\right],$$

$$V_1^{\pi^k}(s_1^k) = \sum_{h=1}^{H}[V_h^{\pi^k}(s_h^k) - V_{h+1}^{\pi^k}(s_{h+1}^k)]$$

$$= \sum_{h=1}^{H} r(s_h^k, a_h^k) + \sum_{h=1}^{H} \left[V_h^{\pi^k}(s_h^k) - r(s_h^k, a_h^k) - [\mathbb{P}V_{h+1}^{\pi^k}](s_h^k, a_h^k)\right]$$

$$+ \sum_{h=1}^{H} \left[[\mathbb{P}V_{h+1}^{\pi^k}](s_h^k, a_h^k) - V_{h+1}^{\pi^k}(s_{h+1}^k)\right]$$

$$= \sum_{h=1}^{H} r(s_h^k, a_h^k) + \sum_{h=1}^{H} \left[[\mathbb{P}V_{h+1}^{\pi^k}](s_h^k, a_h^k) - V_{h+1}^{\pi^k}(s_{h+1}^k)\right],$$

Thus it follows that

$$V_{k,1}(s_1^k) - V_1^{\pi^k}(s_1^k)$$

$$= \sum_{h=1}^{H} \left[V_{k,h}(s_h^k) - r(s_h^k, a_h^k) - [\mathbb{P}V_{k,h+1}](s_h^k, a_h^k)\right]$$

$$+ \sum_{h=1}^{H} \left[[\mathbb{P}V_{k,h+1}](s_h^k, a_h^k) - V_{k,h+1}(s_{h+1}^k)\right] - \sum_{h=1}^{H} \left[[\mathbb{P}V_{h+1}^{\pi^k}](s_h^k, a_h^k) - V_{h+1}^{\pi^k}(s_{h+1}^k)\right]$$

$$= \sum_{h=1}^{H} \left[V_{k,h}(s_h^k) - r(s_h^k, a_h^k) - [\mathbb{P}V_{k,h+1}](s_h^k, a_h^k)\right]$$

$$+ \sum_{h=1}^{H} \left[[\mathbb{P}\check{V}_{k,h+1}](s_h^k, a_h^k) - \check{V}_{k,h+1}(s_{h+1}^k)\right] + \sum_{h=1}^{H} \left[[\mathbb{P}\widetilde{V}_{k,h+1}](s_h^k, a_h^k) - \widetilde{V}_{k,h+1}(s_{h+1}^k)\right].$$

Then we have

$$\sum_{k=1}^{K}\left[V_{k,1}(s_1^k)-V^{\pi^k}(s_1^k)\right]$$

$$\overset{(a)}{\le} |\widetilde{\mathcal{K}}| + 2\sum_{k\in\mathcal{K}}\sum_{h}\min_{l\in[L]}\{1,\beta_{k,l}D_{k,l}(z_{k,h})\} + \sum_{k\in\mathcal{K}}\sum_{h}\left[[\mathbb{P}\check{V}_{k,h+1}](s_h^k,a_h^k) - \check{V}_{k,h+1}(s_{h+1}^k)\right]$$

$$+ \sum_{k\in\mathcal{K}}\sum_{h}\left[[\mathbb{P}\widetilde{V}_{k,h+1}](s_h^k,a_h^k) - \widetilde{V}_{k,h+1}(s_{h+1}^k)\right]$$

$$= 2R + \check{A}_0 + \widetilde{A}_0 + |\widetilde{\mathcal{K}}|,$$

where $(a)$ is due to Lemma C.1. $\qquad\square$

Then we have

$$2R + \check{A}_0 + \widetilde{A}_0 + |\widetilde{\mathcal{K}}|$$

$$\overset{(a)}{\lesssim} R + \sqrt{R} + d_{\mathcal{F}}$$

$$\overset{(b)}{\lesssim} \sqrt{d_{\mathcal{F}}\log N_{\mathcal{F}}\operatorname{Var}_K^*} + \max\{1,c_v\}d_{\mathcal{F}}\log N_{\mathcal{F}}, \qquad (C.26)$$

where $(a)$ holds due to (C.19), (C.21), (C.14), and $(b)$ holds due to (C.23).

Finally, Combining (C.24), (C.25) and (C.26), the high-probability regret bound is given by

$$\operatorname{Regret}(K) = \widetilde{O}\left(\sqrt{d_{\mathcal{F}}\log N_{\mathcal{F}}\operatorname{Var}_K^*} + \max\{1,c_v\}d_{\mathcal{F}}\log N_{\mathcal{F}}\right).$$

$\qquad\square$

# D    MISSING PROOFS IN SECTION C

## D.1    PROOF OF LEMMA C.1

*Proof of Lemma C.1.* On event $\mathcal{E}_k$, for any $l \in [L]$, $f_* \in \mathcal{B}_{k,l}$, it follows that

$$|\widehat{f}_{k,l}(z) - f_*(z)|$$

$$\overset{(a)}{\le} D_{k,l}(z)\sqrt{\sum_{(i,h)\in\Psi_{k,l}} w_{i,h}^2(\widehat{f}_{k,l}(z_{i,h}) - f_*(z_{i,h}))^2 + \sum_{(i,h)\in\widetilde{\Psi}_{k,l}} \widetilde{w}_{i,h}^2(\widehat{f}_{k,l}(\widetilde{z}_{i,h}) - f_*(\widetilde{z}_{i,h}))^2 + \lambda}$$

$$\overset{(b)}{\le} D_{k,l}(z)\sqrt{\beta_{k,l}^2 + \lambda}$$

$$\overset{(c)}{\approx} \beta_{k,l}D_{k,l}(z),$$

where $(a)$ holds due to the definition of $D_{\mathcal{F}}$ in (3.1), $(b)$ holds due to $\sqrt{a+b} \le \sqrt{a} + \sqrt{b}$ for any $a, b \ge 0$, $(c)$ holds due to $\sqrt{\lambda} = O(\beta_{k,l})$ by (C.3).

Furthermore, since this holds for all $l \in [L]$, we have

$$r_h(s_h^k,a_h^k) + [\mathbb{P}V_{k,h+1}](s_h^k,a_h^k)$$

$$= r_h(s_h^k,a_h^k) + f_*(z_{k,h})$$

$$\le \min_{l\in[L]}\{1, r_h(s_h^k,a_h^k) + \widehat{f}_{k,l}(z_{k,h}) + \beta_{k,l}D_{k,l}(z_{k,h})\}$$

$$= V_{k,h}(s_h^k).$$

Thus,

$$V_{k,h}(s_h^k) - r_h(s_h^k, a_h^k) + [\mathbb{P}V_{k,h+1}](s_h^k, a_h^k)$$

$$= \min_{l \in [L]}\{1, r_h(s_h^k, a_h^k) + \widehat{f}_{k,l}(z_{k,h}) + \beta_{k,l}D_{k,l}(z_{k,h})\} - r_h(s_h^k, a_h^k) - f_*(z_{k,h})$$

$$\leq 2\min_{l \in [L]}\{1, \beta_{k,l}D_{k,l}(z_{k,h})\}.$$

Recall the definition of $\bar{\sigma}_{k,h}$ in (5.2), we have for all $l \in [L]$, $\ell \in [L]$,

$$|[\widehat{f}_{k,\ell}(\widetilde{z}_{k,h}) - \widehat{f}_{k,l}^2(z_{k,h})] - [f_*(\widetilde{z}_{k,h}) - f_*^2(z_{k,h})]|$$

$$\leq |\widehat{f}_{k,\ell}(\widetilde{z}_{k,h}) - f_*(\widetilde{z}_{k,h})| + |\widehat{f}_{k,l}^2(z_{k,h}) - f_*^2(z_{k,h})|$$

$$= |\widehat{f}_{k,\ell}(\widetilde{z}_{k,h}) - f_*(\widetilde{z}_{k,h})| + |\widehat{f}_{k,l}(z_{k,h}) + f_*(z_{k,h})| \cdot |\widehat{f}_{k,l}(z_{k,h}) - f_*(z_{k,h})|$$

$$\leq \min\{1, \beta_{k,l}D_{k,\ell}(\widetilde{z}_{k,h})\} + \min\{1, 2\beta_{k,l}D_{k,l}(z_{k,h})\},$$

where the last inequality holds due to $\widehat{f}_{k,l}, f_* \in [0,1]$. Therefore, $[\mathbb{V}V_{k,h+1}](s_h^k, a_h^k)$ is bounded by $\bar{\sigma}_{k,h}^2$:

$$[\mathbb{V}V_{k,h+1}](s_h^k, a_h^k)$$

$$= [\mathbb{P}V_{k,h+1}^2](s_h^k, a_h^k) - [\mathbb{P}V_{k,h+1}]^2(s_h^k, a_h^k) = f_*(\widetilde{z}_{k,h}) - f_*^2(z_{k,h})$$

$$\leq \min_{l \in [L], \ell \in [L]}\left\{1, \widehat{f}_{k,\ell}(\widetilde{z}_{k,h}) - \widehat{f}_{k,l}^2(z_{k,h}) + \min\{1, 2\beta_{k,l}D_{k,l}(z_{k,h})\} + \min\{1, \beta_{k,\ell}D_{k,\ell}(\widetilde{z}_{k,h})\}\right\}$$

$$= \bar{\sigma}_{k,h}^2.$$

Thus

$$\bar{\sigma}_{k,h}^2 - [\mathbb{V}V_{k,h+1}](s_h^k, a_h^k)$$

$$= \min_{l \in [L], \ell \in [L]}\left\{1, \widehat{f}_{k,\ell}(\widetilde{z}_{k,h}) - \widehat{f}_{k,l}^2(z_{k,h}) + \min\{1, 2\beta_{k,l}D_{k,l}(z_{k,h})\} + \min\{1, \beta_{k,\ell}D_{k,\ell}(\widetilde{z}_{k,h})\}\right\}$$

$$\quad - f_*(\widetilde{z}_{k,h}) + f_*^2(z_{k,h})$$

$$\leq 2\min_{l \in [L]}\{1, 2\beta_{k,l}D_{k,l}(z_{k,h})\} + 2\min_{\ell \in [L]}\{1, \beta_{k,\ell}D_{k,\ell}(\widetilde{z}_{k,h})\}.$$

$\square$

## D.2 PROOF OF LEMMA C.2

*Proof of Lemma C.2.* By a union bound, with probability at least $1 - L\delta$, the result follows from Lemma 4.3 using $\{X_{2t-1}, Y_{2t-1}, w_{2t-1}\}_t \cup \{X_{2t}, Y_{2t}, w_{2t}\}_t = \{z_{k,h}, y_{k,h}, w_{k,h}\}_{(k,h) \in \Psi_{K+1,l}} \cup \{\widetilde{z}_{k,h}, y_{k,h}^2, \widetilde{w}_{k,h}\}_{(k,h) \in \widetilde{\Psi}_{K+1,l}}$, $\mathcal{F}$ for $l \in [L]$. We will check the conditions of Lemma 4.3 for all $k \in [K]$ by induction.

First, for $k = 1$, the result holds trivially.

Next, for $k > 1$, suppose event $\bigcap_{i \in [k]} \mathcal{E}_i$ holds, by Lemma C.1, we have for all $(i,h) \in [k] \times [H]$,

$$[\mathbb{V}V_{i,h+1}](s_h^i, a_h^i) \leq \bar{\sigma}_{i,h}^2.$$

Thus from Property 1, for all $l \in [L]$,

$$\mathrm{Var}[y_{i,h}|z_{i,h}] = [\mathbb{V}V_{i,h+1}](s_h^i, a_h^i) \leq \bar{\sigma}_{i,h}^2 \leq 2^l\alpha, \quad w_{i,h}D_{k,h,l}(z_{i,h}) \leq \frac{2^l\alpha}{\gamma} \quad \forall(i,h) \in \Psi_{k+1,l},$$

$$\mathrm{Var}[y_{i,h}^2|\widetilde{z}_{i,h}] = [\mathbb{V}V_{i,h+1}^2](s_h^i, a_h^k) \leq c_v^2\bar{\sigma}_{i,h}^2 \leq 2^l\alpha, \quad \widetilde{w}_{i,h}D_{k,h,l}(\widetilde{z}_{i,h}) \leq \frac{2^l\alpha}{\gamma} \quad \forall(i,h) \in \widetilde{\Psi}_{k+1,l}.$$

Applying Lemma 4.3 with $\sigma = 2^l\alpha$, $\max_{s \in [t]} w_s^2 D_{\mathcal{F}}(X_s; X_{[s-1]}, w_{[s-1]}) = \frac{2^l\alpha}{\gamma}$, we have

$$\sum_{(i,h) \in \Psi_{k+1,l}} w_{i,h}^2(\widehat{f}_{k+1,l}(z_{i,h}) - f_*(z_{i,h}))^2 + \sum_{(i,h) \in \widetilde{\Psi}_{k+1,l}} \widetilde{w}_{i,h}^2(\widehat{f}_{k+1,l}(\widetilde{z}_{i,h}) - f_*(\widetilde{z}_{i,h}))^2 \leq \beta_{k+1,l}^2,$$

that is $f_* \in \mathcal{B}_{k+1,l}$ for all $l \in [L]$, so event $\mathcal{E}_{k+1}$ holds.

Then the proof is completed by induction over $k \in [K]$. $\square$

### D.3 PROOF OF LEMMA C.3

*Proof of Lemma C.3.* We prove the optimism by induction. When $h = H+1$, we have $V_{k,H+1}(\cdot) = V^*_{H+1}(\cdot) = 0$, and the result holds trivially. We assume the statement is true for all $h+1$, and prove the case of $h$. For any $(s,a)$, if $Q_{k,h}(s,a) = 1$, then $Q_{k,h}(s,a) = 1 \geq Q^*_h(s,a)$. Otherwise, we have

$$Q_{k,h}(s,a) - Q^*_h(s,a)$$

$$= \min_{l \in [L]} \left\{ 1, r_h(s,a) + \widehat{f}_{k,l}(s,a,V_{k,h+1}) + \min\{1, \beta_{k,l} D_{k,l}(s,a,V_{k,h+1})\} \right\}$$

$$\qquad - r_h(s,a) - f_*(s,a,V^*_{h+1})$$

$$\geq \min_{l \in [L]} \left( \widehat{f}_{k,l}(s,a,V_{k,h+1}) + \min\{1, \beta_{k,l} D_{k,l}(s,a,V_{k,h+1})\} \right) - f_*(s,a,V_{k,h+1})$$

$$\geq 0,$$

where the first inequality holds due to $V_{k,h+1}(\cdot) \geq V^*_{h+1}(\cdot)$ and the second holds due to Lemma C.1. That is, we have $Q_{k,h}(\cdot,\cdot) \geq Q^*_h(\cdot,\cdot)$ and therefore $V_{k,h}(\cdot) \geq V^*_h(\cdot)$. Then the proof is completed by induction. $\square$

### D.4 PROOF OF LEMMA C.4

*Proof of Lemma C.4.* Recall the definition of $\widetilde{\mathcal{K}}$, we have

$$k \in \widetilde{\mathcal{K}} \iff \exists l \in [L], \sum_{h \in \Psi_{k+1,l} \setminus \Psi_{k,l}} w^2_{k,h} D^2_{k,h,l}(z_{k,h}) + \sum_{h \in \widetilde{\Psi}_{k+1,l} \setminus \widetilde{\Psi}_{k,l}} \widetilde{w}^2_{k,h} D^2_{k,h,l}(\widetilde{z}_{k,h}) > 1.$$

Let $\widetilde{\mathcal{K}}_l$ denote the indices $k$ such that

$$\widetilde{\mathcal{K}}_{l,1} := \left\{ k \in [K] : \sum_{h \in \Psi_{k+1,l} \setminus \Psi_{k,l}} w^2_{k,h} D^2_{k,h,l}(z_{k,h}) + \sum_{h \in \widetilde{\Psi}_{k+1,l} \setminus \widetilde{\Psi}_{k,l}} \widetilde{w}^2_{k,h} D^2_{k,h,l}(\widetilde{z}_{k,h}) > 1 \right\}.$$

Then we have $|\widetilde{\mathcal{K}}| \leq |\bigcup_{l \in [L]} \widetilde{\mathcal{K}}_l| \leq \sum_{l \in [L]} |\widetilde{\mathcal{K}}_l|$. For any $l \in [L]$, we have

$$|\widetilde{\mathcal{K}}_l| \leq \sum_{k=1}^{K} \min \left\{ 1, \sum_{h \in \Psi_{k+1,l} \setminus \Psi_{k,l}} w^2_{k,h} D^2_{k,h,l}(z_{k,h}) + \sum_{h \in \widetilde{\Psi}_{k+1,l} \setminus \widetilde{\Psi}_{k,l}} \widetilde{w}^2_{k,h} D^2_{k,h,l}(\widetilde{z}_{k,h}) \right\}$$

$$\leq \sum_{(k,h) \in \Psi_{K+1,l}} \min \left\{ 1, w^2_{k,h} D^2_{k,h,l}(z_{k,h}) \right\} + \sum_{(k,h) \in \widetilde{\Psi}_{K+1,l}} \min \left\{ 1, \widetilde{w}^2_{k,h} D^2_{k,h,l}(\widetilde{z}_{k,h}) \right\}$$

$$\leq 2 d_{\mathcal{F}}.$$

Taking the summation over $l \in [L]$ gives the upper bound of $|\widetilde{\mathcal{K}}|$. $\square$

### D.5 PROOF OF LEMMA C.5

*Proof of Lemma C.5.* We are to bound $\check{S}_m$ and $\widetilde{S}_m$ with similar arguments.

Recall the definition of $\check{S}_m$ in (C.8), we have

$$\check{S}_m = \sum_{k \in \mathcal{K}} \sum_h [\mathbb{V} \check{V}^{2^m}_{k,h+1}](s^k_h, a^k_h)$$

$$= \sum_{k \in \mathcal{K}} \sum_h \left[ [\mathbb{P} \check{V}^{2^{m+1}}_{k,h+1}](s^k_h, a^k_h) - [\mathbb{P} \check{V}^{2^m}_{k,h+1}]^2(s^k_h, a^k_h)^2 \right]$$

$$= \sum_{k \in \mathcal{K}} \sum_h \left[ [\mathbb{P} \check{V}^{2^{m+1}}_{k,h+1}](s^k_h, a^k_h) - \check{V}^{2^{m+1}}_{k,h+1}(s^k_{h+1}) \right] + \sum_{k \in \mathcal{K}} \sum_h \left[ \check{V}^{2^{m+1}}_{k,h}(s^k_h) - [\mathbb{P} \check{V}^{2^m}_{k,h+1}]^2(s^k_h, a^k_h) \right]$$

$$\qquad + \sum_{k \in \mathcal{K}} \sum_h \left( \check{V}^{2^{m+1}}_{k,h+1}(s^k_{h+1}) - \check{V}^{2^{m+1}}_{k,h}(s^k_h) \right)$$

$$\leq \check{A}_{m+1} + \sum_{k \in \mathcal{K}} \sum_h \left[ \check{V}^{2^{m+1}}_{k,h}(s^k_h) - [\mathbb{P} \check{V}^{2^m}_{k,h+1}]^2(s^k_h, a^k_h) \right]. \tag{D.1}$$

For the second term in (D.1), we have

$$
\sum_{k\in\mathcal{K}}\sum_{h}\left[\check{V}_{k,h}^{2^{m+1}}(s_h^k)-[\mathbb{P}\check{V}_{k,h+1}^{2^m}]^2(s_h^k,a_h^k)\right]
$$

$$
\overset{(a)}{\leq}\sum_{k\in\mathcal{K}}\sum_{h}\left[\check{V}_{k,h}^{2^{m+1}}(s_h^k)-[\mathbb{P}\check{V}_{k,h+1}]^{2^{m+1}}(s_h^k,a_h^k)\right]
$$

$$
=\sum_{k\in\mathcal{K}}\sum_{h}\left[\check{V}_{k,h}(s_h^k)-[\mathbb{P}\check{V}_{k,h+1}](s_h^k,a_h^k)\right]\prod_{i=0}^{m}\left[\check{V}_{k,h}^{2^i}(s_h^k)+[\mathbb{P}\check{V}_{k,h+1}]^{2^i}(s_h^k,a_h^k)\right]
$$

$$
\leq 2^{m+1}\sum_{k\in\mathcal{K}}\sum_{h}\max\left\{\check{V}_{k,h}(s_h^k)-[\mathbb{P}\check{V}_{k,h+1}](s_h^k,a_h^k),0\right\}
$$

$$
\overset{(b)}{\leq}2^{m+1}\sum_{k\in\mathcal{K}}\sum_{h}\max\left\{V_{k,h}(s_h^k)-r(s_h^k,a_h^k)-[\mathbb{P}V_{k,h+1}](s_h^k,a_h^k),0\right\}
$$

$$
\overset{(c)}{\leq}2^{m+1}\sum_{k\in\mathcal{K}}\sum_{h}2\min_{l\in[L]}\{1,\beta_{k,l}\mathcal{D}_{k,l}(z_{k,h})\}
$$

$$
=2^{m+1}\cdot(2R), \tag{D.2}
$$

where $(a)$ holds due to $\mathbb{E}[X^2]\geq(\mathbb{E}[X])^2$, $(b)$ holds due to the definition of $\check{V}_{k,h}$ and $V_h^*(s_h^k)\geq r(s_h^k,a_h^k)+[\mathbb{P}V_{h+1}^*](s_h^k,a_h^k)$, while $(c)$ is due to Lemma C.1. Substituting (D.2) into (D.1), we have

$$
\check{S}_m\leq\check{A}_{m+1}+2^{m+1}\cdot(2R).
$$

Next, we proceed to bound $\widetilde{S}_m$. Recall the definition of $\widetilde{S}_m$ in (C.9), we have

$$
\widetilde{S}_m=\sum_{k\in\mathcal{K}}\sum_{h}[\mathbb{V}\widetilde{V}_{k,h+1}^{2^m}](s_h^k,a_h^k)
$$

$$
=\sum_{k\in\mathcal{K}}\sum_{h}\left[[\mathbb{P}\widetilde{V}_{k,h+1}^{2^{m+1}}](s_h^k,a_h^k)-[\mathbb{P}\widetilde{V}_{k,h+1}^{2^m}]^2(s_h^k,a_h^k)^2\right]
$$

$$
=\sum_{k\in\mathcal{K}}\sum_{h}\left[[\mathbb{P}\widetilde{V}_{k,h+1}^{2^{m+1}}](s_h^k,a_h^k)-\widetilde{V}_{k,h+1}^{2^{m+1}}(s_{h+1}^k)\right]+\sum_{k\in\mathcal{K}}\sum_{h}\left[\widetilde{V}_{k,h}^{2^{m+1}}(s_h^k)-[\mathbb{P}\widetilde{V}_{k,h+1}^{2^m}]^2(s_h^k,a_h^k)\right]
$$

$$
+\sum_{k\in\mathcal{K}}\sum_{h}\left(\widetilde{V}_{k,h+1}^{2^{m+1}}(s_{h+1}^k)-\widetilde{V}_{k,h}^{2^{m+1}}(s_h^k)\right)
$$

$$
\leq\widetilde{A}_{m+1}+\sum_{k\in\mathcal{K}}\sum_{h}\left[\widetilde{V}_{k,h}^{2^{m+1}}(s_h^k)-[\mathbb{P}\widetilde{V}_{k,h+1}^{2^m}]^2(s_h^k,a_h^k)\right]. \tag{D.3}
$$

For the second term in (D.3), we have

$$\sum_{k \in \mathcal{K}} \sum_{h} \left[ \widetilde{V}_{k,h}^{2^{m+1}}(s_h^k) - [\mathbb{P}\widetilde{V}_{k,h+1}^{2^m}]^2(s_h^k, a_h^k) \right]$$

$$\overset{(a)}{\leq} \sum_{k \in \mathcal{K}} \sum_{h} \left[ \widetilde{V}_{k,h}^{2^{m+1}}(s_h^k) - [\mathbb{P}\widetilde{V}_{k,h+1}]^{2^{m+1}}(s_h^k, a_h^k) \right]$$

$$= \sum_{k \in \mathcal{K}} \sum_{h} \left[ \widetilde{V}_{k,h}(s_h^k) - [\mathbb{P}\widetilde{V}_{k,h+1}](s_h^k, a_h^k) \right] \prod_{i=0}^{m} \left[ \widetilde{V}_{k,h}^{2^i}(s_h^k) + [\mathbb{P}\widetilde{V}_{k,h+1}]^{2^i}(s_h^k, a_h^k) \right]$$

$$\leq 2^{m+1} \sum_{k \in \mathcal{K}} \sum_{h} \max \left\{ \widetilde{V}_{k,h}(s_h^k) - [\mathbb{P}\widetilde{V}_{k,h+1}](s_h^k, a_h^k), 0 \right\}$$

$$\overset{(b)}{=} 2^{m+1} \sum_{k \in \mathcal{K}} \sum_{h} \max \left\{ V_h^*(s_h^k) - r(s_h^k, a_h^k) - [\mathbb{P}V_{h+1}^*](s_h^k, a_h^k), 0 \right\}$$

$$\overset{(c)}{\leq} 2^{m+1} \sum_{k \in \mathcal{K}} \sum_{h} \Big[ \max \left\{ V_{k,h}(s_h^k) - r(s_h^k, a_h^k) - [\mathbb{P}V_{k,h+1}](s_h^k, a_h^k), 0 \right\}$$

$$+ |[\mathbb{P}\check{V}_{k,h+1}](s_h^k, a_h^k) - \check{V}_{k,h+1}(s_{h+1}^k)| \Big]$$

$$\overset{(d)}{\leq} 2^{m+1} \sum_{k \in \mathcal{K}} \sum_{h} \left[ 2 \min_{l \in [L]} \{1, \beta_{k,l} D_{k,l}(z_{k,h})\} + |[\mathbb{P}\check{V}_{k,h+1}](s_h^k, a_h^k) - \check{V}_{k,h+1}(s_{h+1}^k)| \right]$$

$$\leq 2^{m+1} \cdot (2R + \check{A}_0), \tag{D.4}$$

where $(a)$ holds due to $\mathbb{E}[X^2] \geq (\mathbb{E}[X])^2$, $(b)$ holds due to the definition of $\widetilde{V}_{k,h}$ and $V_h^{\pi^k}(s_h^k) = r(s_h^k, a_h^k) + [\mathbb{P}V_{h+1}^{\pi^k}](s_h^k, a_h^k)$, $(c)$ holds due to $V_h^*(s_h^k) \geq r(s_h^k, a_h^k) + [\mathbb{P}V_{h+1}^*](s_h^k, a_h^k)$ and the definition of $\check{V}_{k,h}$, while $(d)$ is due to Lemma C.1. Substituting (D.4) into (D.3), we have

$$\widetilde{S}_m \leq \widetilde{A}_{m+1} + 2^{m+1} \cdot (2R + \check{A}_0).$$

$\square$

## D.6 PROOF OF LEMMA C.6

*Proof of Lemma C.6.* Let $X_{k,h} = [\mathbb{P}\check{V}_{k,h+1}^{2^m}](s_h^k, a_h^k) - \check{V}_{k,h+1}^{2^m}(s_{h+1}^k)$, then we have $\mathbb{E}[X_{k,h}|\mathcal{G}_{k,h}] = 0$, $|X_{k,h}| \leq 2$ and $\mathbb{E}[X_{k,h}^2|\mathcal{G}_{k,h}] = [\mathbb{V}\check{V}_{k,h+1}^{2^m}](s_h^k, a_h^k)$. Therefore, for any $m \in \overline{[M]}$, applying variance-aware Freedman's inequality in Lemma E.2, with probability at least $1 - \frac{1}{M+1}\delta$, we have

$$\check{A}_m \leq \sqrt{\zeta \check{S}_m} + \zeta.$$

Thus, taking a union bound over $m \in \overline{[M]}$, with probability at least $1 - \delta$, (C.17) holds. The proofs for (C.18) follow the same arguments as (C.17). $\square$

## D.7 PROOF OF LEMMA C.7

*Proof of Lemma C.7.* On event $\mathcal{E} \cap \mathcal{A}$, we have (C.15) and (C.17) hold by Lemma C.5 and C.6. Substituting the bound of $\check{S}_m$ in (C.15) into (C.17), we have for all $m \in \overline{[M]}$,

$$\check{A}_m \leq \sqrt{\zeta} \cdot \sqrt{\check{A}_{m+1} + 2^{m+1} \cdot (2R)} + \zeta.$$

And we have for all $m \in \overline{[M]}$, $\check{A}_m \leq 2KH$. Then the result follows by Lemma E.6. $\square$

## D.8 PROOF OF LEMMA C.8

*Proof of Lemma C.8.* On event $\mathcal{E} \cap \mathcal{A}$, we have (C.16) and (C.18) hold by Lemma C.5 and C.6. Substituting the bound of $\widetilde{S}_m$ in (C.16) into (C.18), we have for all $m \in \overline{[M]}$,

$$\widetilde{A}_m \leq \sqrt{\zeta} \cdot \sqrt{\widetilde{A}_{m+1} + 2^{m+1} \cdot (2R + \check{A}_0)} + \zeta.$$

And we have for all $m \in \overline{[M]}$, $\widetilde{A}_m \leq 2KH$. Applying Lemma E.6, we have

$$
\begin{aligned}
\widetilde{A}_0 &\leq 2\sqrt{\zeta(2R + \check{A}_0)} + 7\zeta \\
&\overset{(a)}{\leq} 2\sqrt{2\zeta R} + 2\sqrt{\zeta \check{A}_0} + 7\zeta \\
&\overset{(b)}{\leq} 2\sqrt{2\zeta R} + \zeta + \check{A}_0 + 7\zeta \\
&\overset{(c)}{\leq} 4\sqrt{2\zeta R} + 15\zeta,
\end{aligned}
$$

where $(a)$ holds due to $\sqrt{a+b} \leq \sqrt{a} + \sqrt{b}$ for $a, b \geq 0$, $(b)$ holds due to $2\sqrt{ab} \leq a + b$ for $a, b \geq 0$ and $(c)$ holds due to (C.19) in Lemma C.7. $\qquad\square$

### D.9 Proof of Lemma C.9

*Proof of Lemma C.9.* We have for all $k \in \mathcal{K}$, $l \in [L]$,

$$
\sum_{h \in \Psi_{k+1,l} \backslash \Psi_{k,l}} w_{k,h}^2 D_{k,h,l}^2(z_{k,h}) + \sum_{h \in \widetilde{\Psi}_{k+1,l} \backslash \widetilde{\Psi}_{k,l}} \widetilde{w}_{k,h}^2 D_{k,h,l}^2(\widetilde{z}_{k,h}) \leq 1.
$$

By Lemma E.5, it follows that for all $h \in [H]$,

$$
\begin{aligned}
&D_{k,l}(z_{k,h}) \\
&\leq \exp\left\{\frac{1}{2}\left(\sum_{j \in \Psi_{k,h,l} \backslash \Psi_{k,l}} D_{k,j,l}(z_{k,j}) + \sum_{j \in \widetilde{\Psi}_{k,h,l} \backslash \widetilde{\Psi}_{k,l}} D_{k,j,l}(\widetilde{z}_{k,j})\right)\right\} D_{k,h,l}(z_{k,h}) \\
&\leq \exp\left\{\frac{1}{2}\left(\sum_{h \in \Psi_{k+1,l} \backslash \Psi_{k,l}} w_{k,h}^2 D_{k,h,l}^2(z_{k,h}) + \sum_{h \in \widetilde{\Psi}_{k+1,l} \backslash \widetilde{\Psi}_{k,l}} \widetilde{w}_{k,h}^2 D_{k,h,l}^2(\widetilde{z}_{k,h})\right)\right\} D_{k,h,l}(z_{k,h}) \\
&\leq 2 D_{k,h,l}(z_{k,h}).
\end{aligned}
$$

Similarly, for all $k \in \mathcal{K}$, $h \in [H]$, $\ell \in [L]$,

$$
D_{k,\ell}(\widetilde{z}_{k,h}) \leq 2 D_{k,h,\ell}(\widetilde{z}_{k,h}).
$$

Then we have

$$
\begin{aligned}
R &\leq 2 \sum_{k \in \mathcal{K}} \sum_h \min_{l \in [L]}\{1, \beta_{k,l} D_{k,h,l}(z_{k,h})\}, \\
\widetilde{R} &\leq 2 \sum_{k \in \mathcal{K}} \sum_h \min_{\ell \in [L]}\{1, \beta_{k,\ell} D_{k,h,\ell}(\widetilde{z}_{k,h})\}.
\end{aligned}
$$

Setting $\alpha = \sqrt{\frac{d_{\mathcal{F}} \log N_{\mathcal{F}}}{T}}$, applying Lemma E.3 to $R$ and $\widetilde{R}$, we have

$$
R = \widetilde{O}\left(\sqrt{d_{\mathcal{F}} \log N_{\mathcal{F}} \sum_{k \in \mathcal{K}} \sum_h \bar{\sigma}_{k,h}^2} + d_{\mathcal{F}} \log N_{\mathcal{F}}\right), \tag{D.5}
$$

$$
\widetilde{R} = \widetilde{O}\left(c_v \sqrt{d_{\mathcal{F}} \log N_{\mathcal{F}} \sum_{k \in \mathcal{K}} \sum_h \bar{\sigma}_{k,h}^2} + d_{\mathcal{F}} \log N_{\mathcal{F}}\right). \tag{D.6}
$$

Furthermore, according to Lemma C.1, we have

$$
\begin{aligned}
&\sum_{k \in \mathcal{K}} \sum_h \bar{\sigma}_{k,h}^2 \\
&\leq \sum_{k \in \mathcal{K}} \sum_h \left([\mathbb{V} V_{k,h+1}](s_h^k, a_h^k) + 2 \min_{l \in [L]}\{1, 2\beta_{k,l} D_{k,l}(z_{k,h})\} + 2 \min_{\ell \in [L]}\{1, \beta_{k,\ell} D_{k,\ell}(\widetilde{z}_{k,h})\}\right) \\
&\leq S_0 + 4R + 2\widetilde{R}. \tag{D.7}
\end{aligned}
$$

So we have

$$
\begin{aligned}
\widetilde{R} &\stackrel{(D.6)}{\lesssim} c_v \sqrt{d_{\mathcal{F}} \log N_{\mathcal{F}} \sum_{k \in \mathcal{K}} \sum_h \bar{\sigma}_{k,h}^2} + d_{\mathcal{F}} \log N_{\mathcal{F}} \\
&\stackrel{(D.7)}{\lesssim} c_v \sqrt{d_{\mathcal{F}} \log N_{\mathcal{F}} \left(S_0 + R + \widetilde{R}\right)} + d_{\mathcal{F}} \log N_{\mathcal{F}} \\
&\lesssim c_v \sqrt{d_{\mathcal{F}} \log N_{\mathcal{F}} \left(S_0 + R\right)} + \max\{1, c_v^2\} d_{\mathcal{F}} \log N_{\mathcal{F}},
\end{aligned}
\tag{D.8}
$$

where the last inequality holds since $x \le a\sqrt{x} + b$ implies $x \le a^2 + 2b$ for any $x \ge 0$.

And thus

$$
\begin{aligned}
R &\stackrel{(D.5)}{\lesssim} \sqrt{d_{\mathcal{F}} \log N_{\mathcal{F}} \sum_{k \in \mathcal{K}} \sum_h \bar{\sigma}_{k,h}^2} + d_{\mathcal{F}} \log N_{\mathcal{F}} \\
&\stackrel{(D.7)}{\lesssim} \sqrt{d_{\mathcal{F}} \log N_{\mathcal{F}} \left(S_0 + R + \widetilde{R}\right)} + d_{\mathcal{F}} \log N_{\mathcal{F}} \\
&\lesssim \sqrt{d_{\mathcal{F}} \log N_{\mathcal{F}}(S_0 + R)} + d_{\mathcal{F}} \log N_{\mathcal{F}} + \sqrt{d_{\mathcal{F}} \log N_{\mathcal{F}} \widetilde{R}},
\end{aligned}
$$

where the last inequality holds due to $\sqrt{a + b} \le \sqrt{a} + \sqrt{b}$ for any $a, b \ge 0$. Here

$$
\begin{aligned}
&\sqrt{d_{\mathcal{F}} \log N_{\mathcal{F}} \widetilde{R}} \\
&= \sqrt{\max\{1, c_v\} d_{\mathcal{F}} \log N_{\mathcal{F}} \cdot \frac{1}{\max\{1, c_v\}} \widetilde{R}} \\
&\lesssim \sqrt{d_{\mathcal{F}} \log N_{\mathcal{F}} \left(S_0 + R\right)} + \max\{1, c_v\} d_{\mathcal{F}} \log N_{\mathcal{F}},
\end{aligned}
$$

where the inequality holds due to Cauchy-Schwartz inequality and (D.8). Plugin back, we have

$$
\begin{aligned}
R &\lesssim \sqrt{d_{\mathcal{F}} \log N_{\mathcal{F}} \left(S_0 + R\right)} + \max\{1, c_v\} d_{\mathcal{F}} \log N_{\mathcal{F}} \\
&\stackrel{(a)}{\lesssim} \sqrt{d_{\mathcal{F}} \log N_{\mathcal{F}}(\mathrm{Var}_K^* + \check{S}_0 + R)} + \max\{1, c_v\} d_{\mathcal{F}} \log N_{\mathcal{F}} \\
&\stackrel{(C.15)}{\lesssim} \sqrt{d_{\mathcal{F}} \log N_{\mathcal{F}}(\mathrm{Var}_K^* + \check{A}_1 + R)} + \max\{1, c_v\} d_{\mathcal{F}} \log N_{\mathcal{F}} \\
&\stackrel{(C.20)}{\lesssim} \sqrt{d_{\mathcal{F}} \log N_{\mathcal{F}}(\mathrm{Var}_K^* + \sqrt{R} + R)} + \max\{1, c_v\} d_{\mathcal{F}} \log N_{\mathcal{F}} \\
&\lesssim \sqrt{d_{\mathcal{F}} \log N_{\mathcal{F}} \mathrm{Var}_K^*} + \max\{1, c_v\} d_{\mathcal{F}} \log N_{\mathcal{F}},
\end{aligned}
$$

where the $(a)$ holds due to $\mathrm{Var}[X + Y] \le 2 \mathrm{Var}[X] + 2 \mathrm{Var}[Y]$. $\qquad \square$

# E AUXILIARY LEMMAS

**Lemma E.1** (Variance-aware and range-aware Freedman's inequality, Corollary 2 in Agarwal et al. (2023))**.** Let $M \ge m > 0, V \ge v > 0$ be fixed constants, and $\{X_s\}_{s \in [t]}$ be a stochastic process adapted to the filtration $\{\mathcal{G}_s\}_{s \in [t]}$, such that $X_s$ is $\mathcal{G}_s$-measurable. Suppose $\mathbb{E}[X_s | \mathcal{G}_{s-1}] = 0, |X_s| \le M$ and $\sum_{s=1}^t \mathbb{E}[X_s^2 | \mathcal{G}_{s-1}] \le V^2$ almost surely. Then for any $\delta > 0$, with probability at least $1 - (\log(V^2/v^2) + 2)(\log(M/m) + 2)\delta$, we have

$$
\sum_{s=1}^t X_s \le \sqrt{2 \left(2 \sum_{s=1}^t \mathbb{E}[X_s^2 | \mathcal{G}_{s-1}] + v^2\right) \log \frac{1}{\delta}} + \frac{2}{3} \left(2 \max_{s \in [t]} |X_s| + m\right) \log \frac{1}{\delta}.
$$

**Lemma E.2** (Variance-aware Freedman's inequality)**.** Let $M > 0$ be fixed constants, and $\{X_s\}_{s \in [t]}$ be a stochastic process adapted to the filtration $\{\mathcal{G}_s\}_{s \in [t]}$, such that $X_s$ is $\mathcal{G}_s$-measurable. Suppose $\mathbb{E}[X_s | \mathcal{G}_{s-1}] = 0$ and $|X_s| \le M$ almost surely. Then for any $\delta > 0$, with probability at least $1 - 2(\log t + 2)\delta$, we have

$$
\sum_{s=1}^t X_s \le 2 \sqrt{\sum_{s=1}^t \mathbb{E}[X_s^2 | \mathcal{G}_{s-1}] \log \frac{1}{\delta}} + 4M \log \frac{1}{\delta}.
$$

*Proof.* The result follows by applying Lemma E.1 with $V^2 = M^2 t, m = v = M$. $\qquad\square$

**Lemma E.3.** Let $R$, $\alpha$, $\gamma = R\sqrt{\log N_{\mathcal{F}}}$, $L = \lceil \log_2 \frac{R}{\alpha} \rceil$, $\beta_{t,l} = \widetilde{O}(2^l \alpha \sqrt{\log N_{\mathcal{F}}})$. For $t \in [T]$, let disjoint sets $\{\Psi_{t+1,l}\}_{l \in \overline{[L]}}$ be constructed according to ADALEVEL with $D_{t,l} = D_{t,l}(X_t)$. Here, $D_{t,l}(X_t) := D_{\mathcal{F}}(X_t; X_{\Psi_{t,l}}, w_{\Psi_{t,l}})$, where $D_{\mathcal{F}}$ is defined in (3.1). Then we have

$$\sum_{t \in [T]} \min_{l \in [L]}\{R, \beta_{t,l} D_{t,l}(X_t)\} = \widetilde{O}\Big(\sqrt{d_{\mathcal{F}} \log N_{\mathcal{F}} \sum_{t \in [T]} \bar\sigma_t^2} + Rd_{\mathcal{F}} \log N_{\mathcal{F}} + \frac{\alpha^2 T}{\gamma}\sqrt{\log N_{\mathcal{F}}}\Big).$$

Furthermore, setting $\alpha = R\sqrt{\frac{d_{\mathcal{F}} \log N_{\mathcal{F}}}{T}}$ yields

$$\sum_{t \in [T]} \min_{l \in [L]}\{R, \beta_{t,l} D_{t,l}(X_t)\} = \widetilde{O}\Big(\sqrt{d_{\mathcal{F}} \log N_{\mathcal{F}} \sum_{t \in [T]} \bar\sigma_t^2} + Rd_{\mathcal{F}} \log N_{\mathcal{F}}\Big).$$

*Proof.* According to Lemma E.4, we have for all $l \in [L]$,

$$\sum_{t \in \Psi_{T+1,l}} \min\{1, w_t^2 D_{t,l}^2(X_t)\} = O\Big(\dim_{\mathcal{F}}\Big(\sqrt{\lambda/T}\Big) \log T \log(T/\lambda)\Big) = \widetilde{O}(d_{\mathcal{F}}). \qquad (\text{E.1})$$

For all $t \in [T]$, Property 1 holds true. Next, we decompose $[T] = \bigcup_{l \in \overline{[L]}} \Psi_{T+1,l}$ and carefully bound summations within each level.

**Level $l = 0$** For any $t \in \Psi_{T+1,0}$, $D_{t,1}(X_t) \le \frac{2\alpha}{\gamma}$, $\beta_{t,1} = \widetilde{O}(2\alpha\sqrt{\log N_{\mathcal{F}}})$, therefore

$$\sum_{t \in \Psi_{T+1,0}} \min_{l \in [L]}\{R, \beta_{t,l} D_{t,l}(X_t)\} \le \sum_{t \in \Psi_{T+1,0}} \beta_{t,1} D_{t,1}(X_t)$$

$$\lesssim T \cdot 2\alpha\sqrt{\log N_{\mathcal{F}}} \cdot \frac{2\alpha}{\gamma}$$

$$= \widetilde{O}\Big(\frac{\alpha^2 T}{R}\Big).$$

**Level $l = L$** We decompose $\Psi_{T+1,L} = \mathcal{J}_{L,1} \cup \mathcal{J}_{L,2}$ where

$$\mathcal{J}_{L,1} := \Big\{ t \in \Psi_{T+1,L} : w_t = \frac{2^L \alpha}{\gamma D_{t,L}(X_t)} \Big\}, \quad \mathcal{J}_{L,2} := \{ t \in \Psi_{T+1,L} : w_t = 1\}.$$

Thus

$$\sum_{t \in \Psi_{T+1,L}} \min_{l \in [L]}\{R, \beta_{t,l} D_{t,l}(X_t)\} \le R|\mathcal{J}_{L,1}| + \sum_{t \in \mathcal{J}_{L,2}} \beta_{t,L} D_{t,L}(X_t).$$

**Summation over $\mathcal{J}_{L,1}$** For any $t \in \mathcal{J}_{L,1}$, $w_t D_{t,L}(X_t) = \frac{2^L \alpha}{\gamma} \ge \frac{R}{\gamma}$. Thus $1 \le \frac{\gamma^2}{R^2} w_t^2 D_{t,L}^2(X_t)$. Then we have

$$R|\mathcal{J}_{L,1}| \le R \sum_{t \in \mathcal{J}_{L,1}} \frac{\gamma^2}{R^2} w_t^2 D_{t,L}^2(X_t)$$

$$= \frac{\gamma^2}{R} \sum_{t \in \mathcal{J}_{L,1}} \min\{1, w_t^2 D_{t,L}^2(X_t)\}$$

$$\overset{(\text{E.1})}{=} \widetilde{O}\Big(Rd_{\mathcal{F}} \log N_{\mathcal{F}}\Big).$$

**Summation over $\mathcal{J}_{L,2}$** For any $t \in \mathcal{J}_{L,2}$,

$$\beta_{t,L} = \widetilde{O}(2^L \alpha \sqrt{\log N_{\mathcal{F}}}) \lesssim \widetilde{O}(2\bar\sigma_t \log N_{\mathcal{F}}),$$

$$D_{t,L} \le \frac{2^L \alpha}{\gamma}.$$

Thus

$$\sum_{t \in \mathcal{J}_{L,2}} \beta_{t,L} D_{t,L}(X_t) \lesssim \sum_{t \in \mathcal{J}_{L,2}} \bar{\sigma}_t \sqrt{\log N_{\mathcal{F}}} D_{t,L}(X_t)$$

$$\lesssim \sqrt{\log N_{\mathcal{F}}} \cdot \sqrt{\sum_{t \in \mathcal{J}_{L,2}} \bar{\sigma}_t^2} \cdot \sqrt{\sum_{t \in \mathcal{J}_{L,2}} \min\{1, w_t^2 D_{t,L}^2(X_t)\}}$$

$$\overset{(E.1)}{=} \widetilde{O}\Big(\sqrt{d_{\mathcal{F}} \log N_{\mathcal{F}} \sum_{t \in \mathcal{J}_{L,2}} \bar{\sigma}_t^2}\Big).$$

**Level $l \in [L-1]$**   We decompose $\Psi_{T+1,l} = \mathcal{J}_{l,1} \cup \mathcal{J}_{l,2}$ where

$$\mathcal{J}_{l,1} := \Big\{t \in \Psi_{T+1,l} : w_t = \frac{2^l \alpha}{\gamma D_{t,l}(X_t)}\Big\}, \quad \mathcal{J}_{l,2} := \{t \in \Psi_{T+1,l} : w_t = 1\}.$$

Thus

$$\sum_{t \in \Psi_{T+1,l}} \min_{l \in [L]}\{R, \beta_{t,l} D_{t,l}(X_t)\} \leq \sum_{t \in \mathcal{J}_{l,1}} \beta_{t,l+1} D_{t,l+1}(X_t) + \sum_{t \in \mathcal{J}_{l,2}} \beta_{t,l} D_{t,l}(X_t).$$

**Summation over $\mathcal{J}_{l,1}$**   For any $t \in \mathcal{J}_{l,1}$,

$$\beta_{t,l+1} = \widetilde{O}(2^{l+1} \alpha \sqrt{\log N_{\mathcal{F}}}) = \widetilde{O}(4\gamma w_t D_{t,l}(X_t) \sqrt{\log N_{\mathcal{F}}})$$

$$D_{t,l+1}(X_t) \leq \frac{2^{l+1} \alpha}{\gamma} = 2w_t D_{t,l}(X_t).$$

Thus

$$\sum_{t \in \mathcal{J}_{l,1}} \beta_{t,l+1} D_{t,l+1}(X_t) \lesssim \sum_{t \in \mathcal{J}_{l,1}} \gamma w_t D_{t,l}(X_t) \sqrt{\log N_{\mathcal{F}}} \cdot w_t D_{t,l}(x)$$

$$\approx \gamma \sqrt{\log N_{\mathcal{F}}} \sum_{t \in \mathcal{J}_{l,1}} \min\{1, w_t^2 D_{t,l}^2(X_t)\}$$

$$\overset{(E.4)}{=} \widetilde{O}(Rd_{\mathcal{F}} \log N_{\mathcal{F}}).$$

**Summation over $\mathcal{J}_{l,2}$**   For any $t \in \mathcal{J}_{l,2}$,

$$\beta_{t,l} = \widetilde{O}(2^l \alpha \sqrt{\log N_{\mathcal{F}}}) \lesssim \widetilde{O}(2\bar{\sigma}_t \log N_{\mathcal{F}}),$$

$$D_{t,l} \leq \frac{2^l \alpha}{\gamma}.$$

Thus

$$\sum_{t \in \mathcal{J}_{l,2}} \beta_{t,l} D_{t,l}(X_t) \lesssim \sum_{t \in \mathcal{J}_{l,2}} \bar{\sigma}_t \sqrt{\log N_{\mathcal{F}}} D_{t,l}(X_t)$$

$$\lesssim \sqrt{\log N_{\mathcal{F}}} \cdot \sqrt{\sum_{t \in \mathcal{J}_{l,2}} \bar{\sigma}_t^2} \cdot \sqrt{\sum_{t \in \mathcal{J}_{l,2}} \min\{1, w_t^2 D_{t,l}^2(X_t)\}}$$

$$= \widetilde{O}\Big(\sqrt{d_{\mathcal{F}} \log N_{\mathcal{F}} \sum_{t \in \mathcal{J}_{l,2}} \bar{\sigma}_t^2}\Big).$$

Now we put pieces together:

$$\sum_{t \in [T]} \min_{l \in [L]}\{R, \beta_{t,l} D_{t,l}(X_t)\} = \sum_{l \in [L]} \sum_{t \in \Psi_{T+1,l}} \min_{l' \in [L]}\{R, \beta_{t,l'} D_{t,l'}(X_t)\}$$

$$= \widetilde{O}\Big(\sqrt{L d_{\mathcal{F}} \log N_{\mathcal{F}} \sum_{l \in [L]} \sum_{t \in \mathcal{J}_{l,2}} \bar{\sigma}_t^2} + L R d_{\mathcal{F}} \log N_{\mathcal{F}} + \frac{\alpha^2 T}{\gamma} \sqrt{\log N_{\mathcal{F}}}\Big)$$

$$= \widetilde{O}\Big(\sqrt{d_{\mathcal{F}} \log N_{\mathcal{F}} \sum_{t \in [T]} \bar{\sigma}_t^2} + R d_{\mathcal{F}} \log N_{\mathcal{F}} + \frac{\alpha^2 T}{\gamma} \sqrt{\log N_{\mathcal{F}}}\Big).$$

$\square$

**Lemma E.4** (Lemma D.6 in Jia et al. (2024))**.** Let $D_{\mathcal{F}}$ be defined in (3.1), and $w_t \in [0, 1]$ for all $t \in [T]$. Then we have

$$\sum_{t \in [T]} \min\{1, w_t^2 D_{\mathcal{F}}(X_t; X_{[t-1]}, w_{[t-1]})\} = O\Big( \dim_{\mathcal{F}} \left( \sqrt{\lambda/T} \right) \log T \log(T/\lambda) \Big).$$

**Lemma E.5** (Lemma H.4 in Huang et al. (2024))**.** Let $D_{\mathcal{F}}$ be defined in (3.1). Then for any $t > t_0 \geq 1$, we have

$$D_{\mathcal{F}}^2(X_t; X_{[t_0]}, w_{[t_0]}) \leq \exp\left\{ \sum_{s=t_0+1}^{t-1} w_s^2 D_{\mathcal{F}}^2(X_s; X_{[s-1]}, w_{[s-1]}) \right\} D_{\mathcal{F}}^2(X_t; X_{[t-1]}, w_{[t-1]}).$$

**Lemma E.6** (Lemma H.6 in Huang et al. (2024))**.** Let $\lambda_1, \lambda_2, \lambda_4 > 0$, $\lambda_3 \geq 1$ and $i' = \lceil \log_2 \lambda_1 \rceil$. Let $a_0, a_1, a_2, \ldots, a_{i'}$ be non-negative reals such that $a_i \leq \lambda_1$ for any $0 \leq i \leq i'$, and $a_i \leq \lambda_2 \sqrt{a_{i+1} + 2^{i+1} \cdot \lambda_3} + \lambda_4$ for any $0 \leq i < i'$. Then we have

$$a_0 \leq \max\left\{ \left( \lambda_2 + \sqrt{\lambda_2^2 + \lambda_4} \right)^2, \lambda_2 \sqrt{4\lambda_3} + \lambda_4 \right\} \leq \lambda_2 \sqrt{4\lambda_3} + 4\lambda_2^2 + 3\lambda_4,$$

$$a_1 \leq \max\left\{ \left( \lambda_2 + \sqrt{\lambda_2^2 + \lambda_4} \right)^2, \lambda_2 \sqrt{8\lambda_3} + \lambda_4 \right\} \leq \lambda_2 \sqrt{8\lambda_3} + 4\lambda_2^2 + 3\lambda_4.$$

