# OpenReview forum: "Multi-Level Regression for Nonlinear Contextual Bandits and RL: Second-order and Horizon-free Regret Bounds"
_ICLR.cc/2026/Conference — Submitted to ICLR 2026_

### Official Review · Reviewer_xoVH · 2025-10-29

**Soundness:** 3
**Presentation:** 2
**Contribution:** 3
**Rating:** 4
**Confidence:** 3

**Summary:**

The paper proposes a multi-level regression (MLR) framework (ADALEVEL + weighted regressions) for nonlinear contextual bandits with unknown, heteroscedastic variance, yielding the algorithm UCB-MLR. The key idea is to partition data by both uncertainty and estimated variance, run separate regressions per level, and aggregate optimistic UCBs. Under standard realizability plus an auxiliary assumption on the squared-reward model, the paper proves a second-order regret, matching prior lower bounds in the main term. The framework is extended to model-based RL, giving a horizon-free, second-order regret bound under an additional variance-of-variance assumption.

**Strengths:**

A good theoretical contribution:

1. Close a know gap: main term scales as $\sqrt{d_F}$, matching the lower bound in bandit setting.

2. Use a regression oracle and shows how to compute the uncertainty measure $D_F$ via binary search with $\tilde{O}(1)$ oracle calls.

**Weaknesses:**

1. Lack of empirical studies. Even though this paper is a theoretical work, I still think it is proposing a novel algorithm. That means the feasibility and practical evidence are critical to include. I don't think the paper needs lots of simulation and experiments but some experiment to show the performance and show the theoretical insight (to guide the practical use case) are required to be a satisfied paper.

2. Subsequence from above point, since there is no practical results, I don't see how to choose or set the (hyper)parameters for the algorithms. I am concerned about the feasibility. For example, the choice of $\alpha$, $\tilde{\alpha}$, $gamma$, $\tilde{gamma}$ and constant $R$ and confidence scales $\beta_{t, \ell}$ etc. For theory, parameters are fully specified in the appendix (but some require unknown complexity terms). A short “practical tuning” section would improve usability.

3. RL extension assumptions: the RL bound adds a variance-of-variance condition (and deterministic rewards), which may be restrictive; more concrete families where this holds would help.

**Questions:**

1. Why using the non-standard assumption for $y_t^2$? Please provide concrete distributional families (beyond sub-Gaussian) where $Var[y^2|X] \leq c_V^2 R^2 Var[y|x]$ holds, and examples where it fails. How sensitive are your bounds/algorithms to violations?

2. In Eq 4.2, there is $g \in F$ but earlier introduce a distinct class. I assume it is a typo? or do you intend to state $F = G$ in the analysis? If $G \neq F$, how do $d_G$, $\log N_G$ enter computation and tuning?

3. What concrete F families admit the weighted-regression oracle in near-linear time and allow efficient evaluation of $D_F$ via binary search (e.g., RKHS with kernel ridge, GLMs)? Any hardness results if $F$ is a deep net class?

4. For the RL extension, please restate the exact variance-relation assumption and give examples (e.g., linear mixture MDPs with bounded features) where it holds; can you weaken it without losing horizon-free behavior?

5. Lemma 4.3 uses a covering-net union bound with weights bounded by $W$. Under ADALEVEL, do we always have $W \leq 1$? Please point to the exact place in property 1 that enforces this bound.

---

> ### Author Response · Authors · 2025-11-21
> **Official Comment by Authors 1/2**
>
> We thank the reviewer for their thoughtful assessment and for recognizing our contribution in closing the known gap for nonlinear contextual bandits. We address your specific questions below.
>
> **Q1&2** Lack of empirical studies and concerns regarding hyperparameter tuning.
>
> **A1&2** Our work primarily focuses on theoretical analysis, and serves as a first step to use the multi-level regression scheme for nonlinear function approximation. We provide rigorous proofs demonstrating effectiveness of our algorithm. We believe the multi-level regression technique offers valuable insights for practical use. Developing an efficient, practical version of the algorithm is an important question for future work. As noted in Tables 1 and 2, the state-of-the-art baselines, e.g., [1-4], are also purely theoretical works. We follow this established line of inquiry to provide the first rigorous proofs for this setting.
>
> Regarding hyperparameters, e.g., confidence scales $\beta$, it is standard in theoretical literature to define them based on complexity measures $d_\mathcal{F}, \log N_\mathcal{F}$ to guarantee regret bounds. While practical implementations often use heuristics to tune these as calculating Eluder dimensions is hard, deriving such heuristics requires extensive empirical study that is outside the scope of this theoretical framework. We view our work as providing the foundational theoretical insight the reviewer requested, paving the way for future applied work.
>
> ---
>
> **Q3** RL extension assumptions: the RL bound adds a variance-of-variance condition (and deterministic rewards), which may be restrictive; more concrete families where this holds would help.\
> **Q4** Why use the non-standard assumption for $y_t^2$? Please provide concrete distributional families (beyond sub-Gaussian) where $\text{Var}[y^2] \le c_v^2 R^2 \text{Var}[y]$ holds, and examples where it fails. How sensitive are your bounds/algorithms to violations?\
> **Q7** For the RL extension, please restate the exact variance-relation assumption and give examples (e.g., linear mixture MDPs with bounded features) where it holds; can you weaken it without losing horizon-free behavior?
>
> **A3&4&7** The variance-relation assumption for RL is described in Assumption 3.5. We clarify that this is not a restrictive assumption. As detailed in our Common Response QA1, for any reward function bounded by $R$, this inequality holds automatically with $c_v \le 2$. Thus, we can drop this term and the corresponding assumptions from our results.
> We assume deterministic rewards in the RL extension primarily for clean presentation. This is a common convention in recent theoretical RL literature. Generalizing to stochastic rewards is straightforward by adding a standard martingale concentration argument (see [5] for an analysis of heavy-tailed rewards with function approximation), but it adds notation without changing the core instance-dependent logic.
>
> ---
>
> **Q5** Typo in Eq 4.2. How do $d_\mathcal{G}, \log N_\mathcal{G}$ enter computation and tuning?
>
> **A5** Yes, this is a typo. The regression should minimize over $g \in \mathcal{G}$. We will correct this in the revision. Regarding computation, we assume access to a regression oracle for $\mathcal{G}$, similar to $\mathcal{F}$. See Common Response QA2 for the relationship between these classes.
>
> [1] Zhao, Heyang, et al. "Variance-dependent regret bounds for linear bandits and reinforcement learning: Adaptivity and computational efficiency." The Thirty Sixth Annual Conference on Learning Theory. PMLR, 2023.\
> [2] Wang, Kaiwen, et al. "More Benefits of Being Distributional: Second-Order Bounds for Reinforcement Learning." International Conference on Machine Learning. PMLR, 2024.\
> [3] Huang, Jiayi, et al. "Horizon-free and instance-dependent regret bounds for reinforcement learning with general function approximation." International Conference on Artificial Intelligence and Statistics. PMLR, 2024.\
> [4] Wang, Zhiyong, et al. "Model-based RL as a Minimalist Approach to Horizon-Free and Second-Order Bounds." The Thirteenth International Conference on Learning Representations (2024).\
> [5] Huang, Jiayi, et al. "Tackling heavy-tailed rewards in reinforcement learning with function approximation: Minimax optimal and instance-dependent regret bounds." Advances in Neural Information Processing Systems 36 (2023): 56576-56588.\
> [6] Li, Yunfan, et al. "Low-switching policy gradient with exploration via online sensitivity sampling." International Conference on Machine Learning. PMLR, 2023.\
> [7] Ye, Chenlu, et al. "Corruption-robust offline reinforcement learning with general function approximation." Advances in Neural Information Processing Systems 36 (2023): 36208-36221.

---

> > ### Author Response · Authors · 2025-11-21
> > **Official Comment by Authors 2/2**
> >
> > **Q6** What concrete F families admit the weighted-regression oracle in near-linear time and allow efficient evaluation of $D_F$ via binary search (e.g., RKHS with kernel ridge, GLMs)? Any hardness results if $F$ is a deep net class?
> >
> > **A6** The regression oracle assumption is standard in general function approximation literature [3, 6, 7]. For Linear Mixture Models, the oracle admits an efficient closed-form solution [1]. For Convex Function Classes, [3, 6] demonstrate that the uncertainty $D_\mathcal{F}$ can be efficiently evaluated via binary search. While finding the global optimum for deep neural networks is theoretically NP-hard (a known hardness result), in practice, gradient-based optimization is effectively used as an approximate oracle. Empirically, prior work such as [7] has successfully approximated this uncertainty by computing the standard deviation of a network ensemble, supporting the practical viability of the weighting technique.
> >
> > ---
> >
> > **Q8** Lemma 4.3 uses a covering-net union bound with weights bounded by $W$. Under ADALEVEL, do we always have $W \le 1$? Please point to the exact place in Property 1 that enforces this bound.
> >
> > **A8** Yes, $W \le 1$ holds by construction for all levels $l \in [L]$. In Algorithm 2 (ADALEVEL), Line 6 sets the weight $w_t = \frac{2^{l} \alpha}{\gamma D_{t,l}(X_t)}$. A data point is assigned to level $l$ (Line 1) only if $\gamma D_{t,l}(X_t) > 2^l \alpha$, implying the denominator is larger than the numerator. Consequently, whenever the weight formula is applied, $w_t < 1$. If the condition is not met (the "else" branch), we explicitly set $w_t = 1$. Thus, the weight is always bounded by 1.
> >
> > [1] Zhao, Heyang, et al. "Variance-dependent regret bounds for linear bandits and reinforcement learning: Adaptivity and computational efficiency." The Thirty Sixth Annual Conference on Learning Theory. PMLR, 2023.\
> > [2] Wang, Kaiwen, et al. "More Benefits of Being Distributional: Second-Order Bounds for Reinforcement Learning." International Conference on Machine Learning. PMLR, 2024.\
> > [3] Huang, Jiayi, et al. "Horizon-free and instance-dependent regret bounds for reinforcement learning with general function approximation." International Conference on Artificial Intelligence and Statistics. PMLR, 2024.\
> > [4] Wang, Zhiyong, et al. "Model-based RL as a Minimalist Approach to Horizon-Free and Second-Order Bounds." The Thirteenth International Conference on Learning Representations (2024).\
> > [5] Huang, Jiayi, et al. "Tackling heavy-tailed rewards in reinforcement learning with function approximation: Minimax optimal and instance-dependent regret bounds." Advances in Neural Information Processing Systems 36 (2023): 56576-56588.\
> > [6] Li, Yunfan, et al. "Low-switching policy gradient with exploration via online sensitivity sampling." International Conference on Machine Learning. PMLR, 2023.\
> > [7] Ye, Chenlu, et al. "Corruption-robust offline reinforcement learning with general function approximation." Advances in Neural Information Processing Systems 36 (2023): 36208-36221.

---

### Official Review · Reviewer_rGUp · 2025-10-31

**Soundness:** 3
**Presentation:** 3
**Contribution:** 3
**Rating:** 6
**Confidence:** 3

**Summary:**

In this paper, the authors propose a multi-level regression framework for nonlinear contextual bandits and model-based RL. The key algorithmic idea is ADALEVEL, which partitions data using both a per-point uncertainty proxy and an online-learned variance upper bound; separate weighted regressions are run per level and actions are chosen by the minimum across level-wise UCBs. A central technical ingredient is a Bernstein/Freedman-style concentration lemma for nonlinear regression that decouples variance from uncertainty, fixing the $\sqrt{d_F}$ gap that persisted in prior nonlinear multi-layer analyses. For bandits, UCB-MLR achieves a second-order regret
$\tilde O(\sqrt{d_F \log N_F \sum_t \sigma_t^2} + \max(1,C) R d_F \log N_F)$
with $C=\max(1,c_v)\sqrt{(d_G \log N_G)/(d_F \log N_F)}$, matching the known lower bound in the main term under standard realizability. For RL, the same multi-level idea combined with value-targeted regression yields an instance-dependent, horizon-free bound
$\tilde O(\sqrt{d_F \log N_F \mathrm{Var}^\*_K} + \max(1,c_v) d_F \log N_F)$.

**Strengths:**

The paper studies nonlinear contextual bandits with unknown variances, where prior multi-layer methods incurred an extra $\sqrt{d_F}$ in the leading term. The decoupled Bernstein bound together with variance-aware leveling removes this and reaches the minimax-tight second-order rate, resolving a standing gap and aligning with the spirit of the linear case. The adaptive leveling idea is clean and potentially reusable in other online learning settings. The paper is clearly written and the approach is easy to follow.

**Weaknesses:**

The lower-order term in the bandit regret depends on the second-moment modeling through $C$. When $d_G \log N_G \gg d_F \log N_F$, this term scales like
$\tilde O(R \cdot \max(1,c_v)\sqrt{(d_F \log N_F)(d_G \log N_G)})$
rather than purely $d_F \log N_F$. While I do not have a concrete instance where $d_G \log N_G$ dominates, this dependence is a side effect of learning the variance via a separate class and may leave a (lower-order) gap in unfavorable modelings.

Minor typos and suggestions:
- In Equation (4.2), the squared-target regression for $g$ should minimize over $\mathcal G$ (not $\mathcal F$).
- In Line 8 of Algorithm 1, to align with Assumption 3.2, ADALEVEL should receive $c_v R \bar \sigma_t$ (or state $R=1$ after normalization).
- The paper uses both $l$ and $\ell$ as level indices for the $\mathcal F$- and $\mathcal G$-branches; consider renaming one of them for readability.

**Questions:**

- What is the exact definition of $d_P$ when comparing to distributional/transition-model baselines? It would be helpful to state this explicitly in the introduction alongside Tables 1.
- The paper assumes the second moment is realizable. While common in recent work, is this assumption necessary here? In RL, could one leverage the structure of $V^2$ directly (e.g., via clipped/Catoni-style targets) to avoid a separate modeling burden while keeping horizon-free rates?

---

> ### Author Response · Authors · 2025-11-21
>
> We thank the reviewer for the encouraging assessment and for recognizing that our work resolves the standing gap in nonlinear contextual bandits to achieve the minimax-tight second-order rate. We address your specific questions below.
>
> **Q1** The lower-order term in the bandit regret depends on the second-moment modeling through $C$. When $d_\mathcal{G} \log N_\mathcal{G} \gg d_\mathcal{F}\log N_\mathcal{F}$, this term scales like $\tilde{O}(R \max\\{1,c\\} \sqrt{d_\mathcal{F} \log N_\mathcal{F} d_\mathcal{G} \log N_\mathcal{G}})$ rather than purely $d_\mathcal{F} \log N_\mathcal{F}$. This dependence is a side effect of learning the variance via a separate class and may leave a (lower-order) gap in unfavorable modelings.\
> **Q4** The paper assumes the second moment is realizable. While common in recent work, is this assumption necessary here? In RL, could one leverage the structure of $V^2$ directly (e.g., via clipped/Catoni-style targets) to avoid a separate modeling burden while keeping horizon-free rates?
>
> **A1&4**
> - For contextual bandits, we acknowledge the additional assumption of $\mathcal{G}$ for general function approximation. While this introduces a dependence on $d_\mathcal{G}$ in the lower-order term, we emphasize that this assumption is strictly weaker than assuming realizability of the full reward distribution $\mathcal{P}$ as in [1], since $\mathcal{G}$ can be automatically induced by $\mathcal{P}$. Furthermore, as detailed in our Common Response QA2, the complexity of moments is upper-bounded by the complexity of the distribution. Consequently, our lower-order term satisfies $R \sqrt{d_\mathcal{F} \log N_\mathcal{F} d_\mathcal{G} \log N_\mathcal{G}} \le R d_\mathcal{P} \log N_\mathcal{P}$, matching the result of [1] under their assumptions while remaining valid in broader settings where full distributional realizability may not hold.
> - In the model-based RL literature, the realizability assumption on the transition model $\mathcal{P}$ is standard. Actually, we do not require access to the full transition distribution as required by [2]. We only assume the first two moments of the next-state value function are realizable via $\mathcal{F}$, which is naturally induced by $\mathcal{P}$. Therefore, we do not require an additional assumption for the second-order moment in RL.
> - While Catoni-style estimators [3] effectively handle heavy-tailed rewards, they do not inherently provide the local variance estimates required for the Bernstein-style bound. To achieve second-order and horizon-free rates, we must leverage the information of variance, which requires explicit estimation of the second-order moment rather than just robust mean estimation. We capture the model by conducting a simple least-squared regression targeted on the next-state value functions and squared value functions in a coherent manner.
>
> ---
>
> **Q2** Minor typos and suggestions.
>
> **A2** We thank the reviewer for the careful read. We will incorporate all suggested fixes in the revision:
> - We will correct Eq (4.2) to minimize over $g \in \mathcal{G}$.
> - We will update Algorithm 1 (Line 8) to explicitly state the input as $c_v \bar{\sigma}_t$.
> - We will rename the level indices (e.g., using $k$ or another notation for the $\mathcal{G}$-branch) to improve readability and avoid confusion between the two regression branches.
>
> ---
>
> **Q3** What is the exact definition of $d_P$ when comparing to distributional/transition-model baselines? It would be helpful to state this explicitly in the introduction alongside Table 1.
>
> **A3** We will add the explicit definition in the revision. $d_\mathcal{P}$ refers to the distributional Eluder dimension or Hellinger Eluder dimension of the reward distribution class $\mathcal{P}$, as introduced in [1]. It measures the complexity of learning the full distribution rather than just the mean. As noted in QA1 above and our Common Response QA2, while $d_\mathcal{P}$ is generally larger than $d_\mathcal{F}$, we establish the relationship that estimating distributions is harder than estimating moments, justifying our tighter dependencies in settings where full distributional learning is unnecessary.
>
> [1] Wang, Kaiwen, et al. "More Benefits of Being Distributional: Second-Order Bounds for Reinforcement Learning." International Conference on Machine Learning. PMLR, 2024.\
> [2] Wang, Zhiyong, et al. "Model-based RL as a Minimalist Approach to Horizon-Free and Second-Order Bounds." The Thirteenth International Conference on Learning Representations (2024).\
> [3] Ye, Chenlu, et al. "Catoni contextual bandits are robust to heavy-tailed rewards." International Conference on Machine Learning. PMLR, 2025.

---

### Official Review · Reviewer_WtmF · 2025-10-31

**Soundness:** 3
**Presentation:** 3
**Contribution:** 2
**Rating:** 2
**Confidence:** 3

**Summary:**

This paper proposes a unified Multi-Level Regression (MLR) framework for both contextual bandits and model-based reinforcement learning with general function approximation. The framework leverages multi-level uncertainty partitioning and weighted regression to jointly estimate rewards and higher-order variance information. The authors claim three main advantages: (1) instance-dependent regret bounds, (2) second-order exploration guarantees, and (3) horizon-free regret in RL settings. The paper demonstrates theoretical benefits compared to prior work such as Huang et al. (2024) and Wang et al. (2025), particularly achieving improved computational tractability while maintaining strong statistical guarantees. Regret analyses, confidence bounds, and computational complexity are rigorously addressed.

**Strengths:**

Theoretical generality and unification
The proposed multi-level regression framework is conceptually appealing and unifies several strands of variance-aware learning in contextual bandits and RL under a single analytical blueprint. The results meaningfully extend horizon-free and instance-dependent learning beyond linear mixture MDPs.

Improvement over closely related approaches
The paper offers a more favorable combination of assumptions, regret guarantees, and computational efficiency compared to:
Huang et al. (2024): avoids suboptimal dependencies on higher-order value moments
O-MBRL (Wang et al., 2025): avoids requiring access to the full transition distribution
The work achieves a strong balance between theory and algorithmic feasibility.

**Weaknesses:**

1.Strong realizability assumptions limit practicality
The requirement that both the mean reward function \*f\* and the variance model \*g\* lie in known hypothesis classes is quite restrictive—especially in model-based RL, where model misspecification often induces compounding errors. The feasibility of estimating higher-order variance surrogates in realistic environments is not adequately addressed.

2.Insufficient explanation of the multi-level mechanism
The ADALEVEL component and level partitioning strategy play a central role but lack conceptual intuition. It remains unclear: why this particular partitioning is necessary, how level granularity influences performance, what happens under noisy or misclassified uncertainty estimates. The methodological novelty therefore feels underspecified.

3.Lack of empirical evaluation
The submission provides no experiments to validate computational advantages or regret improvements in practice. For a venue like ICLR, this severely weakens the impact—especially given that comparable works demonstrate sample-efficient performance empirically.

4.Limited discussion of highly related prior research
While the paper cites key works (Ye et al., 2025; Huang et al., 2024; Zhao et al., 2023), the comparative analysis is mostly surface-level. It would be beneficial to more explicitly quantify: the precise statistical gaps closed relative to each approach, the behavior of MLR under approximate function classes.

5.Exposition challenges
Presentation is dense and relies heavily on appendix proofs. Several key insights are obscured by technical details, making the core contribution harder to appreciate for general RL readers.

This paper follows previous works problem, lacking problem novelty.

**Questions:**

see weaknesses.

---

> ### Author Response · Authors · 2025-11-21
> **Official Comment by Authors 1/2**
>
> We thank the reviewer for their time and for acknowledging the theoretical generality and unification of our framework. We appreciate the feedback and address the specific concerns below.
>
> **Q1** Strong realizability assumptions limit practicality.
>
> **A1** We respectfully clarify that our realizability assumptions are actually weaker than those in comparable state-of-the-art works.
> - For contextual bandits, while [1] assume realizability of the full reward distribution $\mathcal{P}$, we only assume realizability of the first two moments $\mathcal{F}$ and $\mathcal{G}$, which are automatically induced by $\mathcal{P}$. Thus, our assumption is strictly weaker. Furthermore, as detailed in our Common Response QA2, our regret bound matches that of [1] when $\mathcal{P}$ has a finite Eluder dimension and covering number.
> - For RL, the realizability assumption on transition model class is standard in model-based RL literature. Actually, we do not require access to the transition model class $\mathcal{P}$ as required by [2]. We only assume the first two moments of the next-state value function are realizable via $\mathcal{F}$, which is naturally induced by $\mathcal{P}$. Therefore, we do not require an additional assumption for the second-order moment in RL. See Assumption 3.4 and its discussion.
> - Regarding misspecification, we can always allow misspecification errors which will show up in the final regret bound additively. Such analyses are usually routine in RL theory and often omitted when the additional complication introduced is orthogonal to the main technical insights of the paper. In fact, all of the closely related works we compare with assume realizability and do not consider misspecification errors.
>
> ---
>
> **Q2** Insufficient explanation of the multi-level mechanism.
>
> **A2** The core innovation of ADALEVEL is to partition data based on both uncertainty and variance instead of just uncertainty. By separating data points with high variance from those with low variance, the algorithm can run separate regressions. This allows us to apply tighter confidence bounds to the low-variance levels, preventing high-variance outliers from loose bounds that degrade the overall learning rate. This adaptive leveling technique allows for adaptive, instance-dependent learning. We have provided the high-level idea in Section 4.1, and established rigorous proof for its correctness. See Property 1 for properties of ADALEVEL, and Appendix B for a detailed explanation and proof.
>
> ---
>
> **Q3** Lack of empirical evaluation.
>
> **A3** Our work primarily focuses on theoretical analysis, and serves as a first step to use the multi-level regression scheme for nonlinear function approximation. We provide rigorous proofs demonstrating effectiveness of our algorithm. We believe the multi-level regression technique offers valuable insights for practical use. Developing an efficient, practical version of the algorithm is an important question for future work. While ICLR started as a deep-learning-focused conference, it has now grown to include many more topics, including hardcore theory, in a way nearly indistinguishable from NeurIPS/ICML. Many RL theory papers without empirical components have been published in ICLR and later proved to be influential. The reviewer mentioned that “comparable works demonstrate … empirically”, but all the closely related works we compare with in Tables 1 and 2, e.g., Ye et al., (2025), Huang et al., (2024) and Zhao et al., (2023), are purely theoretical.
>
> ---
>
> **Q4** Limited discussion of highly related prior research (Ye et al., 2025; Huang et al., 2024; Zhao et al., 2023).
>
> **A4** We believe we have provided a detailed quantitative comparison. Tables 1 and 2 explicitly list the regret bounds of our results alongside state-of-the-art works, e.g., Ye et al. (2025), Huang et al. (2024), and Zhao et al. (2023), highlighting the precise statistical gaps such as dependence on $d_\mathcal{F}$ and handling nonlinear function approximation. We additionally highlight our contributions compared to their works in Related Work and Proof Sketch sections in the main text.
>
> [1] Wang, Kaiwen, et al. "More Benefits of Being Distributional: Second-Order Bounds for Reinforcement Learning." International Conference on Machine Learning. PMLR, 2024.\
> [2] Wang, Zhiyong, et al. "Model-based RL as a Minimalist Approach to Horizon-Free and Second-Order Bounds." The Thirteenth International Conference on Learning Representations (2024).

---

> > ### Author Response · Authors · 2025-11-21
> > **Official Comment by Authors 2/2**
> >
> > **Q5** Exposition challenges.
> >
> > **A5** Indeed, we have established rigorous proofs of our theoretical results in the appendix, which is very common for theoretical papers that appear in top venues such as ICLR.
> >
> > ---
> >
> > **Q6** This paper follows previous works problem, lacking problem novelty.
> >
> > **A6** While the problem setting (contextual bandits and RL) has been established, our solution addresses a specific, unsolved challenge: designing an algorithm for nonlinear contextual bandits that achieves minimax-optimal, second-order regret under standard realizability assumptions. Previous works either suffered from suboptimal dependence on complexity parameters or required stronger distributional assumptions. Bridging this gap requires the novel algorithmic techniques (MLR) we introduced.

---

### Official Review · Reviewer_xHti · 2025-11-06

**Soundness:** 2
**Presentation:** 2
**Contribution:** 3
**Rating:** 4
**Confidence:** 4

**Summary:**

This submission claims to shave a $\sqrt{d}$ factor in the variance-dependent regret bound for learning bandits and episodic MDPs with general function approximation under various additional assumptions, especially the known uniform upper bound on the reward variance. The proposed algorithm employ an algorithmic peeling technique for both the second and forth order moments, whose analysis relies on a newly established martingale concentration for nonlinear online regression with uniform noise upper bound.

**Strengths:**

- The proof is well-written.
- This is the first work under general function approximation claiming to be able to nearly recover the tight dimension dependency for linear bandits and linear mixture MDPs. And the recovery of Zhao et al. (2023) in the MDP setting is faithful and correct.

**Weaknesses:**

- The outline from Lemma 4.3 to Theorem 4.2 is strictly weaker than Zhao et al. (2023) in the linear case because Zhao et al. (2023) does not need the uniform upper bound $\sup_{t} \sigma_t$ to be known to the agent.
- Table 1 is misleading. Actually, Zhao et al. does not need this assumption involving $c_v$, which means to have an apple-to-apple comparison, the second term in the authors' regret bound (for linear bandits) should scale with $R^2$ instead of $R$.
- The authors omit the dependency w.r.t. $\mathcal{G}$ here. But it is crucial to notice that $\mathcal{G}$ does NOT have an apple-to-apple counterpart in Zhao et al. (2023). Since $g_* \in \mathcal{G}$ essentially models $f_*^2(x_t) + \mathrm{Var}[\epsilon_t | x_t]$, and $\mathrm{Var}(\epsilon_t|x_t)$ can have very complex dependency w.r.t. $x_t$, which is allowed and is NOT modeled using an additional Eluder dimension in Zhao et al. (2023), the regret bound in this table might be considered misleading or an overclaim, unless the authors can propose a concrete upper bound of $d_\mathcal{G}$ in the setting of Zhao et al. (2023)

**Questions:**

- In the MDP setting, if the authors' total reward is bounded by $1$ in each episode, the $c_v$ in Assumption 3.5 can be just $c_v=2$ following the Lemma E.7 in https://arxiv.org/pdf/2407.15007, right?
- In the MDP setting, since the authors are only peeling the second and the forth moments, is it possible to follow the spirit of [The proof of Lemma 3.2 in https://arxiv.org/pdf/2407.15007] to bypass the tedious high-order expansion arguments in Appendix C.2 and Appendix D?

---

> ### Author Response · Authors · 2025-11-21
>
> We thank the reviewer for their detailed comments and for acknowledging our proofs as well-written and our recovery of [1] as faithful. We address the specific concerns below.
>
> **Q1** Does the analysis from Lemma 4.3 to Theorem 4.2 require the uniform upper bound of variance to be known, thereby making it weaker than [1]?
>
> **A1** No, our algorithm does not require the uniform upper bound of variance to be known to the agent. As stated in the abstract and introduction, we operate strictly under the setting where the variance is _unknown_ to the agent. While the theoretical analysis utilizes upper bounds, the algorithm itself relies on the estimated variance $\bar{\sigma}_t$, which is learned adaptively from the data. The known variance case mentioned by the reviewer refers to a simplified setting which has been well studied in previous works, e.g., [2], and is not the focus of our contribution.
>
> ---
>
> **Q2** Table 1 is misleading. The paper needs additional assumptions involving $c_v$ compared to [1] and the $c_v$-related term in the authors' regret bound should be larger in Table 1.\
> **Q4** In the MDP setting, if the authors' total reward is bounded by $1$ in each episode, then in Assumption 3.5 can be just $c_v=2$ following the Lemma E.7 in [4], right?
>
> **A2&4** We agree with the reviewer that for any reward distribution bounded by $R$ (or normalized to $1$), the constant $c_v$ satisfies $c_v \le 2$. So we can drop this term and the corresponding assumptions from our results. See Common Response QA1 for details.
>
> ---
>
> **Q3** In Table 1, the authors omit the dependency w.r.t. $\mathcal{G}$, which can be very complex and is unnecessary in [1]. The regret bound in this table might be considered misleading or an overclaim.
>
> **A3** We acknowledge that we omitted the explicit dependency on $\mathcal{G}$ for brevity, and we will add a column specifying realizability assumptions in Table 1 for clarity in the revision. We admit the additional realizability assumption of $\mathcal{G}$ for contextual bandits, since we study nontrivial nonlinear generalization compared to [1]. Furthermore, compared to [3], who assume realizability of the full reward distribution $\mathcal{P}$, our assumption is strictly weaker as we only require realizability of the first two moments, $\mathcal{F}$ and $\mathcal{G}$, which are automatically induced by $\mathcal{P}$. Furthermore, our regret bound can be proved to match that of [3] when $\mathcal{P}$ has finite Eluder dimension and covering number. See Common Response QA2 for details.
>
> ---
>
> **Q5** In the MDP setting, since the authors are only peeling the second and the forth moments, is it possible to follow the spirit of The proof of Lemma 3.2 in [4] to bypass the tedious high-order expansion arguments in Appendix C.2 and Appendix D?
>
> **A5** While this is an insightful suggestion, there is a fundamental difference in the problem settings. [4] primarily address Imitation Learning (IL), where the data distribution is often more stable. Our work addresses RL, which involves the difficult trade-off between exploration and exploitation. To achieve a horizon-free regret bound in RL using Value-Targeted Regression (VTR), we must rigorously control the error propagation through recursive Bellman updates. The high-order expansion analysis in Appendix C.2 is specifically designed to peel off these propagated errors across the planning horizon, a complexity not typically present in IL.
>
> [1] Zhao, Heyang, et al. "Variance-dependent regret bounds for linear bandits and reinforcement learning: Adaptivity and computational efficiency." The Thirty Sixth Annual Conference on Learning Theory. PMLR, 2023.\
> [2] Jia, Zeyu, et al. "How does variance shape the regret in contextual bandits?." Advances in Neural Information Processing Systems 37 (2024): 83730-83785.\
> [3] Wang, Kaiwen, et al. "More Benefits of Being Distributional: Second-Order Bounds for Reinforcement Learning." International Conference on Machine Learning. PMLR, 2024.\
> [4] Foster, Dylan J., Adam Block, and Dipendra Misra. "Is behavior cloning all you need? understanding horizon in imitation learning." Advances in Neural Information Processing Systems 37 (2024): 120602-120666.

---

> ### Comment · Reviewer_xHti · 2025-11-21
>
> Thank you for the response and acknowledgement.
>
> However, the first response is incorrect because:
> - from the algorithmic perspective, the design of $\bar{\sigma}_t^2$ in equation (4.3) requires the uniform $\sigma^2$ as the input;
> - from the perspective of statistical guarantee, the uniform upper bound $\sigma^2$ in Lemma 4.3 for the weighted conditional variance is evident, which makes the guarantee, (**when specialized to the linear setting**), strictly weaker than that Freedman-type inequality in Zhao et al. (2023).

---

> > ### Author Response · Authors · 2025-11-22
> >
> > We thank the reviewer for the follow-up and for closely examining the algorithmic details. We realize there were some miscommunications and misunderstanding and apologize for the ambiguity in our earlier response, offer the following clarifications to address your concerns.
> >
> > ### Algorithmic Perspective (Eq 4.3 in Algorithm 1)
> >
> > First, our algorithm does not require $\sup\_t \sigma\_t$ (which is the tightest possible upper bound, that depends on the nature’s choice of $x\_1, \ldots, x\_T$) to be known to the agent. While we do require $\sigma$ to be known to the agent, **it can be a loose bound of $\sup\_t \sigma\_t$**. In fact, it suffices to set $\sigma = R$ (the reward boundedness) and our guarantee will still hold, and we already implicitly do that in our current proof.
> >
> > In particular, in Line 8 of Algorithm 2 (AdaLevel), to ensure the candidate set $\min\\{l\in[L],l>l_t: \bar{\sigma}_t\le 2^l\alpha\\}$ is non-empty, we set the number of levels $L = \lceil\log_2\frac{R}{\alpha}\rceil$ such that $2^L \alpha \approx R \ge \bar{\sigma}_t$, where the second inequality implicitly uses $\sigma = R$. And crucially, $\sigma$ doesn’t contribute to the regret bound. This is because, as detailed in Eq. B.12 in the Appendix, the estimated variance $\bar{\sigma}_t$ is bounded by the estimated variance term (the second term in Eq 4.3) instead of $\sigma$.
> >
> > That said, we agree that the current writing can lead to the misunderstanding that $\sigma$ may need to be a nontrivial and tight upper bound of variance. To address this, we will explicitly replace $\sigma^2$ with $R^2$ in Eq 4.3 in the revision.
> >
> > ### Statistical Guarantee (Lemma 4.3)
> >
> > We acknowledge Lemma 4.3 (a Bernstein-type inequality) technically utilizes $\sigma$ as an almost-sure upper bound. Whereas the Freedman-type inequality used in Zhao et al., (2023) utilizes the sum of variances and does not require $\sigma$.
> >
> > While the theoretical analysis utilizes upper bounds, the algorithm itself relies on the estimated variance $\bar{\sigma}_t$, which is learned adaptively from the data.
> >
> > The primary contribution of our work is resolving the open problem in the general nonlinear setting. As noted in Table 1 and the Related Work section, previous attempts (e.g., Pacchiano, (2025), Jia et al.,(2024)) generalized the Freedman-type inequality in Zhao et al., (2023) to nonlinear settings but incur a $\sqrt{d_\mathcal{F}}$-suboptimal gap due to nonlinear structural complexities. Our novel approach successfully closes this gap, achieving minimax-optimal, second-order regret in complex nonlinear problems.

---

### Author Response · Authors · 2025-11-21
**Common Response QA1**

We respond to the technical questions raised by multiple reviewers here.

**Q1** Under what conditions does the variance relation $\text{Var}[y^2] \le c_v^2 R^2 \text{Var}[y]$ hold, and how large is the constant $c_v$?

**A1** While we included $c_v$ as a variable parameter in our guarantee, we realized after submission that $c_v \le 2$ always holds for any bounded reward distribution, so we can drop this term and the corresponding assumptions from our results. Specifically, for any random variable $y \in [0, R]$, we have:
$$
\text{Var}[y^2] = \mathbb{E}[(y^2 - \mathbb{E}[y^2])^2] \le \mathbb{E}[(y^2 - \mathbb{E}[y]^2)^2] \le 4R^2 \mathbb{E}[(y - \mathbb{E}[y])^2] = 4R^2 \text{Var}[y].
$$
Here, the first inequality follows from the property that the expectation minimizes the squared error, and the second holds because $|y^2 - \mathbb{E}[y]^2| = |y - \mathbb{E}[y]| \cdot |y + \mathbb{E}[y]| \le 2R |y - \mathbb{E}[y]|$. Consequently, for the variance assumptions in contextual bandits (Assumption 3.2) and RL (Assumption 3.5), we have $c_v \le 2$.
With this constant established, the final regret bound for contextual bandits simplifies to:
$$
\tilde{O}\left(\sqrt{d_\mathcal{F} \log N_\mathcal{F} \sum_{t \in [T]}\sigma_t^2} + \max\left\\{1, \sqrt{\frac{d_\mathcal{G} \log N_\mathcal{G}}{d_\mathcal{F} \log N_\mathcal{F}}}\right\\} R d_\mathcal{F} \log N_\mathcal{F}\right).
$$

---

### Author Response · Authors · 2025-11-21
**Common Response QA2 1/2**

**Q2** Why is the second-order realizability assumption ($g\_* \in \mathcal{G}$) necessary in contextual bandit problems, and how does it relate to previous work?

**A2**
### Necessity of the assumption

To establish tight second-order regret bounds in the nonlinear contextual bandit setting, accurate variance estimation is crucial. Therefore, we require the realizability of the second-order moment ($g_* \in \mathcal{G}$) to tractably estimate the variance.

Eliminating this assumption while maintaining a $\sqrt{d_\mathcal{F}}$-depended regret remains a highly challenging open question, which we leave for important future work.

---

### Author Response · Authors · 2025-11-21
**Common Response QA2 2/2**

### Relationship with prior works

Our assumption is a middle ground bridging linear and fully distributional approaches. While we make stronger assumptions than the linear bandits in [1], our setting addresses the significantly harder problem of general function approximation. Crucially, our assumption is strictly weaker than [2], who assume realizability of the full reward distribution $\mathcal{P}$. In contrast, we only assume realizability of the first two moments, $\mathcal{F}$ and $\mathcal{G}$, which are automatically induced by $\mathcal{P}$. Furthermore, the extra $d_\mathcal{G}$-related term appears only in the lower-order term not scaling with $\sqrt{T}$.

To rigorously demonstrate that our assumption is weaker, we establish a relationship between our complexity measures ($d_\mathcal{F}, N_\mathcal{F}, d_\mathcal{G}, N_\mathcal{G}$) and the distributional complexity ($d_\mathcal{P}, N_\mathcal{G}$). Since $\mathcal{P}$ is a collection of distributions, we consider the Hellinger Eluder dimension.

**Definition (Hellinger Eluder Dimension).** For a model class $\mathcal{P} \subset \mathcal{X} \to \Delta(\mathbb{R})$, let $d_\mathcal{P}(\alpha)$ be the length of the longest sequence $(x_i, P_i, P'_i)$ where $P_i, P'_i \in \mathcal{P}$ satisfy $\sum\_{j<i} H^2(P_i(x_j), P'_i(x_j)) \le \alpha^2$ but $H^2(P_i(x_i), P'_i(x_i)) > \alpha^2$. Here, $H(P, Q)$ denotes the Hellinger distance: $H^2(P,Q) = \frac{1}{2} \int (\sqrt{dP(x)} - \sqrt{dQ(x)})^2$.

Roughly speaking, this replaces Euclidean distance with Hellinger distance in the standard Eluder dimension definition. The moment classes $\mathcal{F}$ and $\mathcal{G}$ are induced from $\mathcal{P}$ as
\begin{aligned}
\mathcal{F} &= \\{f: \mathcal{X} \to \mathbb{R}: | \exists P \in \mathcal{P}, f(x) = \mathbb{E}\_{y \sim P(x)}[y]\\},\\\\
\mathcal{G} &= \\{g: \mathcal{X} \to \mathbb{R}: | \exists P \in \mathcal{P}, g(x) = \mathbb{E}\_{y \sim P(x)}[y^2]\\}.
\end{aligned}

**Core Inequality: Hellinger Dominates Moments**
Hellinger distance controls the distance between moments for bounded variables. Using the relation between Total Variation (TV) and Hellinger distance, $TV(P,Q) \le \sqrt{2} H(P,Q)$, we have:
\begin{aligned}
|f_P - f_Q| &= |\int y (dP - dQ)| \le R \cdot TV(P,Q) \le \sqrt{2}R \cdot H(P,Q),\\\\
|g_P - g_Q| &= |\int y^2 (dP - dQ)| \le R^2 \cdot TV(P,Q) \le \sqrt{2}R^2 \cdot H(P,Q).
\end{aligned}

**Relationship between Covering Numbers**
Since Hellinger distance dominates moment distance, an $\epsilon$-cover of $\mathcal{P}$ induces an $O(\epsilon)$-cover of $\mathcal{F}$ and $\mathcal{G}$. Thus, $\log N_\mathcal{F} \le \log N_\mathcal{P} $ and $\log N_\mathcal{G} \le \log N_\mathcal{P}$.

**Relationship between Eluder Dimensions**
Since the Hellinger distance is a stronger metric, if a sequence of contexts $x_1​,…,x_t​$ is independent with respect to the mean, i.e., we can predict past means well but fail on the current mean, it must also be independent with respect to the Hellinger distance, because failing on the mean implies a large Hellinger distance.

In other words, the complexity of learning the distribution upper-bounds the complexity of learning its moments. Consider two distributions $P_1, P_2$ with the same mean but different variances (e.g., a peak vs. a flat distribution). They are indistinguishable in $\mathcal{F}$ ($f_1(x) = f_2(x)$) but distinguishable in $\mathcal{P}$ ($H(P_1, P_2) > 0$). An algorithm learning $\mathcal{P}$ must resolve this uncertainty, whereas an algorithm learning $\mathcal{F}$ assumes it is resolved. Thus, a sequence of points can be independent with respect to $\mathcal{P}$ while being dependent with respect to $\mathcal{F}$. Consequently, $d_\mathcal{F} \le d_\mathcal{P}$ and $d_\mathcal{G} \le d_\mathcal{P}$.

**Conclusion**
Substituting these relationships into our regret bound (Theorem 4.2), the lower-order term satisfies:
$$
R \sqrt{d_\mathcal{F} \log N_\mathcal{F} d_\mathcal{G} \log N_\mathcal{G}}) \le R d_\mathcal{P} \log N_\mathcal{P}.
$$
This confirms that when $\mathcal{P}$ has finite Eluder dimension and covering number, our bound matches the state-of-the-art distributional result of [2] while remains valid in broader settings where full distributional realizability may not hold.

[1] Zhao, Heyang, et al. "Variance-dependent regret bounds for linear bandits and reinforcement learning: Adaptivity and computational efficiency." The Thirty Sixth Annual Conference on Learning Theory. PMLR, 2023.\
[2] Wang, Kaiwen, et al. "More Benefits of Being Distributional: Second-Order Bounds for Reinforcement Learning." International Conference on Machine Learning. PMLR, 2024.

---

### Meta-Review · Area_Chair_y5Qf · 2026-01-05

**Summary:**

This paper proposes a unified multi-level regression framework for nonlinear contextual bandits and model-based RL with heteroscedastic, unknown variances. Reviewers credit the work for recovering linear-style dimension dependence under general function approximation and for providing computationally plausible oracle-based subroutines.

The main weaknesses are (i) practicality: strong realizability assumptions, especially a realizable variance estimator $g \in \mathcal G$ and the reliance on regression oracles. The other weakness is the comparison with the existing works like Zhao et al. and Wang et al., which makes this paper does not meet the bar of acceptance.

**Reviewer Concerns:**

The authors provided more theoretical justification in their response addressing the questions regarding the assumption of $c_v$. However, there are still a majority of concerns remaining opening.
1. Reviewer xHti mentioned that the algorithm requires the knowledge of $\sigma$. The authors claims that this update can be *approximated* by a heuristic lower bound $\sigma_t$. However, this still does not resolve the question and using an approximated variance cannot lead to the precise bound.
2. Reviewer WtmF mentioned that the realizability assumption, especially $g \in G$ is questionable. The authors acknowledged that they indeed need the variance term $g \in G$ in the bandit setting. Thus this weakness remains unaddressed. Similar question / concern is also raised by Reviewer rGUp as well.

**Reviewer Scores:**

The reviewer score is unlikely to change or at least cannot make significant impact of the decision.

---

### Decision · Program_Chairs · 2026-01-26

Reject